# Favorability of Loss Landscape with Regularization Requires Both Large Overparametrization and Initialization

**Etienne Boursier**                                        *etienne.boursier@inria.fr*
*INRIA*
*LMO, Université Paris-Saclay*
*Orsay, France*

**Matthew Bowditch**                                 *matthew.bowditch@warwick.ac.uk*
*Mathematics Institute*
*University of Warwick*
*Coventry, UK*

**Matthias Englert**                                     *m.englert@warwick.ac.uk*
*Department of Computer Science*
*University of Warwick*
*Coventry, UK*

**Ranko Lazić**                                          *r.s.lazic@warwick.ac.uk*
*Department of Computer Science*
*University of Warwick*
*Coventry, UK*

**Reviewed on OpenReview:** *https://openreview.net/forum?id=jbUOTjjhfg*

## Abstract

The optimization of neural networks under weight decay remains poorly understood from a theoretical standpoint. While weight decay is standard practice in modern training procedures, most theoretical analyses focus on unregularized settings. In this work, we investigate the loss landscape of the $\ell_2$-regularized training loss for two-layer ReLU networks. We show that the landscape becomes favorable – i.e., spurious local minima represent a negligible fraction of local minima – under large overparametrization, specifically when the network width $m$ satisfies $m \gtrsim \min(n^d, 2^n)$, where $n$ is the number of data points and $d$ the input dimension. More precisely in this regime, almost all constant activation regions contain a global minimum and no spurious local minima. We further show that this level of overparametrization is not only sufficient but also necessary via the example of orthogonal data. Finally, we demonstrate that such loss landscape results primarily hold relevance in the large initialization regime. In contrast, for small initializations – corresponding to the feature learning regime – optimization can still converge to spurious local minima, despite the favorability of the landscape.

## 1  Introduction

While the empirical success of machine learning, particularly with overparametrized architectures, has been remarkable across a wide range of tasks (He et al., 2016; Jumper et al., 2021), our theoretical understanding of why these models perform so well remains limited. A large portion of the theoretical literature in machine learning has focused on providing optimization guarantees that ensure convergence to a global minimum of the training loss. While these results offer important foundations, they often rely on strong assumptions and idealized settings, such as smooth activations and infinitely wide architectures (Chizat & Bach, 2018;

Mei et al., 2018; Rotskoff & Vanden-Eijnden, 2022; Wojtowytsch, 2020) or initialization regimes that are not representative of the feature learning happening in practice when training neural networks (Jacot et al., 2018; Du et al., 2019; Arora et al., 2019; Chizat et al., 2019). More specific optimization analyses provide a comprehensive understanding of training dynamics, yet they are frequently restricted to simplified regimes, such as specific data examples (Lyu et al., 2021; Boursier et al., 2022; Glasgow, 2024) or linear architectures (Woodworth et al., 2020), limiting their applicability to realistic models and datasets.

Another line of work shifts focus from dynamics to geometry, aiming to characterize the structure of the loss landscape itself (Sun et al., 2020). These studies investigate conditions under which the landscape is *favorable* – that is, when the non-convexity of the loss does not pose a fundamental obstacle to finding global minima efficiently, and the loss landscape is free from spurious minima that could impede optimization. While the literature suggests that the loss landscape of overparametrized neural networks is often favorable (Laurent & von Brecht, 2018; Kawaguchi, 2016; Venturi et al., 2019), such results do not directly guarantee that standard optimization algorithms will converge to global minima of the training loss. In particular, several works demonstrate the existence of a *descending direction* from any non-minimizing parameter, i.e., a direction along which the training loss does not increase. However, certain spurious local minima may still persist (Safran & Shamir, 2018; Goldblum et al., 2020), where this descending direction corresponds to a flat direction: one in which the loss remains constant rather than decreasing. As a result, common optimization methods can still become trapped in these local minima, hindering convergence.

Boursier & Flammarion (2025a) further demonstrated, through an analysis of training dynamics, that even infinitely wide ReLU networks can converge to spurious local minima of the unregularized loss when initialized with sufficiently small weights. This result highlights that, despite the existence of descent directions and favorable structures in the loss landscape, optimization algorithms may still converge to undesirable stationary points. In particular, convergence to spurious local minima remains possible, even in the presence of descending paths, underscoring a gap between geometric properties of the loss landscape and the practical behavior of optimization dynamics.

While much of the existing literature has provided valuable insights into training without regularization, the importance of regularization cannot be overstated in practice. In the context of (stochastic) gradient descent, $\ell_2$-regularization is equivalent to weight decay. However, this equivalence no longer holds for more modern optimizers, such as Adam (Kingma & Ba, 2015) or Muon (Jordan et al., 2024), where the two approaches can lead to different optimization dynamics. In this work, we focus on $\ell_2$-regularization, and, motivated by gradient descent–type methods, we use the terms $\ell_2$-regularization and weight decay interchangeably.

Weight decay is standard in modern neural network training, playing a crucial role in controlling model complexity and ensuring generalization (Shalev-Shwartz & Ben-David, 2014; Bach, 2017), particularly when training large, overparametrized networks on real-world data. Despite its widespread use, the theoretical understanding of how regularization influences the loss landscape and optimization dynamics remains incomplete. In this work, we study the training dynamics of two-layer ReLU networks with $\ell_2$-regularization. Our analysis begins with a geometric perspective, characterizing the loss landscape induced by weight decay. We then examine how this geometric understanding translates (or fails to translate) into practical optimization behaviors by analyzing the training dynamics on specific data examples.

**Contributions.** After introducing the considered problem and setting in Section 3, we study the regularized loss landscape of overparametrized networks in Section 4. We show that, under large overparametrization – specifically, when the number of parameters $m$ exceeds $\min(2^n, n^d)$, where $n$ is the number of training examples and $d$ is the data dimension – the loss landscape exhibits a favorable structure: most activation regions contain a global minimum of the training loss and are devoid of spurious local minima. This result suggests that, in this overparametrized regime, convergence to global minima is typically straightforward. However, the loss landscape alone does not fully explain the trajectory of optimization. We then study the limitations of this loss landscape analysis, particularly emphasizing its relevance in the large initialization regime. In Section 5, we illustrate through a specific data example that with small initializations (i.e., in the mean field or feature learning regime), the parameters may converge to interpolators with suboptimal $\ell_2$-norm, but minimizing instead the number of non-zero neurons. Furthermore, in Section 6, we explore the case of orthogonal data and demonstrate that the large overparametrization $m \gtrsim \min(2^n, n^d)$ is not merely a

sufficient condition for a favorable loss landscape, but is also necessary to guarantee convergence to a global minimum of the regularized loss, regardless of the initialization regime. Finally, we experimentally illustrate our different findings in Section 7, providing concrete examples where our theoretical results are validated and offering a deeper understanding of the optimization dynamics in various regimes.

At a high level, our work underscores the additional challenges introduced when incorporating a regularization term into the training loss. Specifically, we find that a significantly larger overparametrization is required to ensure convergence to a global minimum, both from a landscape perspective and from an optimization perspective, especially in the case of orthogonal data. Furthermore, we also highlight the limitations of relying solely on the loss landscape analysis: while it provides valuable insights in the large initialization regime, this picture becomes more complex in the small initialization regime. In this case, the implicit bias of optimization plays a critical role, potentially guiding the optimization process towards aspurious local minimum that presents other regularities such as model sparsity.

## 2 Related work

**No spurious valley and mode connectivity.** Early studies of loss landscapes focused on identifying settings in which the loss landscape is *benign*, i.e., when all local minima are global (Laurent & von Brecht, 2018; Kawaguchi, 2016). However, once ReLU activations are introduced into the architecture, spurious local minima become prevalent (Safran & Shamir, 2018; Yun et al., 2019; Goldblum et al., 2020; He et al., 2020).

Despite the existence of such spurious minima, the loss landscape of ReLU networks retains some favorable geometric properties. In the regime of mild overparametrization – i.e., when the number of hidden units exceeds the number of training samples – Haeffele & Vidal (2017); Venturi et al. (2019) indeed showed that there is no bad valley for the unregularized loss. That is, from any point in the parameter space, there exists a continuous path along which the training loss does not increase and that leads to a global minimum. Moreover, under this same overparametrization, the set of global minima (and more generally, all sublevel sets) is connected (Nguyen, 2019; Sharif-Nassab et al., 2020; Nguyen, 2021; Nguyen et al., 2021; Simsek et al., 2021), a property commonly referred to as *mode connectivity* (Garipov et al., 2018; Draxler et al., 2018).

Building on convex reformulations of the training objective for two-layer ReLU networks, Wang et al. (2022); Kim et al. (2025) extended these results to the regularized setting. More precisely they demonstrated that, with an appropriate (albeit non-explicit) level of overparametrization, there exists no bad valley and the set of global minima of the $\ell_2$-regularized loss remains connected. Although such results do not preclude the existence of spurious local minima, they do imply that no spurious *strict* local minimum exists.

**Loss landscape through activation patterns.** A characteristic feature of ReLU networks is that the parameter space can be partitioned into distinct regions, determined by the training data, such that the network behaves as a linear model within each region. Each of these regions corresponds to a fixed activation pattern of the hidden neurons, creating a piecewise-linear structure in the loss landscape. This piecewise-linear partitioning underpins the convex reformulations proposed by Pilanci & Ergen (2020); Ergen & Pilanci (2021); Wang et al. (2022); Mishkin & Pilanci (2023), which enabled the characterization of mode connectivity for the $\ell_2$-regularized loss (Kim et al., 2025). It has also served as a foundation in numerous optimization studies (Boursier et al., 2022; Chistikov et al., 2023; Min et al., 2024; Boursier & Flammarion, 2025b), where the training dynamics become more tractable when analyzed within a fixed activation pattern.

While the total number of global and local minima is infinite, the number of distinct activation regions is finite for a given dataset and fixed network width. Leveraging this observation, Karhadkar et al. (2024) introduces a novel perspective on the unregularized loss landscape: rather than focusing on individual critical points, they analyze the distribution of constant activation regions that contain global minima versus those containing spurious local minima. They show that, under mild overparametrization, regions containing spurious local minima are relatively rare. In the special case of univariate input data, they further demonstrate that most activation regions contain at least one global minimum. Our first contribution builds upon this approach by incorporating an $\ell_2$ regularization term into the analysis. We extend the characterization of the loss landscape to the regularized setting and provide more general results regarding the proportion of activation regions that contain global minima.

## 3 Setting

We consider a two-layer ReLU network parametrized by $\theta := (W, a) \in \mathbb{R}^{m \times (d+1)}$ that corresponds to the function $f_\theta$ defined for any $x \in \mathbb{R}^d$ as

$$f_\theta(x) = a^\top \sigma(Wx), \tag{1}$$

where $W \in \mathbb{R}^{m \times d}$ corresponds to the inner layer of the network, $a \in \mathbb{R}^m$ is the output layer and the ReLU activation $\sigma : z \mapsto \max(0, z)$ is applied component wise. Additionally, we write $w_i \in \mathbb{R}^d$ for the $i$-th row of the matrix $W$. With training data $(x_k, y_k)_{k \in [n]} \in \mathbb{R}^{(d+1) \times n}$, we consider the following regularized regression problem

$$\min_\theta \frac{1}{n} \sum_{k=1}^n (f_\theta(x_k) - y_k)^2 + \lambda \|\theta\|_2^2 \tag{Reg-$\lambda$}$$

where $\lambda > 0$ is the regularization parameter. When interpolation is possible (e.g. if $m \geq n$), solutions of Equation (Reg-$\lambda$) converge towards solutions of the following problem as $\lambda \to 0$:

$$\min_\theta \|\theta\|_2^2 \text{ s.t. } f_\theta(x_k) = y_k \text{ for any } k \in [n]. \tag{min-norm}$$

A key aspect of the ReLU activation is that it is piecewise linear. As a consequence, Ergen & Pilanci (2021) proposed an equivalent convex problem to both Equations (Reg-$\lambda$) and (min-norm). This equivalent problem is designed via a partitioning of the parameter space into cones where the output function $f_\theta$ behaves linearly in both $W$ and $a$ inside each of these cones. Formally, we associate to each binary matrix $A \in \{0, 1\}^{m \times (n+1)}$ the *activation cone* $\mathcal{C}^A$ given by

$$\mathcal{C}^A := \left\{ (W, a) \in \mathbb{R}^{m \times (d+1)} \mid \forall (i, k) \in [m] \times [n], \mathbb{1}(w_i^\top x_k \geq 0) = A_{i,k} \text{ and } \mathbb{1}(a_i \geq 0) = A_{i,n+1} \right\}. \tag{2}$$

Concretely, activation cones partition the parameter space into regions where all activations are fixed (determined by the signs of $w_i^\top x_k$ for each neuron $w_i$ and datapoint $x_k$). Within each cone, the network is linear in the parameters – though not jointly linear in $(W, a)$ – which makes these regions particularly useful for both landscape and dynamics analysis.

Note that some activation cones might be empty sets. These activation cones play a key role in the optimization of two-layer ReLU networks. Notably, they allow for the formulation of convex problems equivalent to both Equations (Reg-$\lambda$) and (min-norm), yielding insightful conclusions on their global minima (Ergen & Pilanci, 2021; Wang et al., 2022). They also are crucial in understanding the training dynamics of two-layer ReLU networks, as all neurons with the same activation (i.e., row $A_i$) follow the same dynamics (up to rescaling) (Maennel et al., 2018).

### 3.1 Notations

We write $e_1, \ldots, e_d$ for the standard basis of $\mathbb{R}^d$, and $\mathbb{S}^{d-1}$ for the $d$-dimensional unit sphere. For a set $\mathcal{C}$, we denote its closure by $\bar{\mathcal{C}}$. We note $f(t) = \mathcal{O}(g(t))$, if there exists a constant $C$ such that for any $t$, $|f(t)| \leq Cg(t)$. Similarly we note $f(t) = \Omega(g(t))$, if there exists a constant $C > 0$ such that $f(t) \geq Cg(t)$.

## 4 Overparametrization and favorable loss landscape

Our goal is to show that, under sufficient overparametrization, the loss landscape is favorable in the sense that most local minima are global. However, the set of all local minima is infinite and difficult to characterize directly. To make this phenomenon more tractable, we focus on activation cones, which provide a natural partition of the parameter space and, crucially, are finite in number. Karhadkar et al. (2024) study the unregularized problem, corresponding to Equation (Reg-$\lambda$) with $\lambda = 0$, by analyzing the number of non-spurious cones. They show, when the model is mildly overparametrized ($m \gtrsim n/d$), that only a small fraction of activation cones contain differentiable spurious stationary points. Theorem 1 below provides a similar picture in the presence of regularization ($\lambda > 0$), stating that in the case of large overparametrization, nearly all activation cones do not contain spurious local minima. Additionally, we show that nearly all activation cones also contain global minima, which is another strong argument in favor of the *favorability of the loss landscape.*

**Theorem 1.** *Let $\varepsilon \in (0,1)$. If $m = \Omega\left(\min(n^d, 2^n)\log(\frac{n}{\varepsilon})\right)$, then for any $\lambda > 0$, in all except at most an $\varepsilon$ fraction of non-empty activation cones $\mathcal{C}^A$ it simultaneously holds:*

> *(i) the activation cone $\overline{\mathcal{C}}^A$ contains a global minimum of Equation (Reg-$\lambda$) (respectively Equation (min-norm));*

> *(ii) the activation cone $\overline{\mathcal{C}}^A$ does not contain any spurious local minimum of Equation (Reg-$\lambda$) restricted to $\overline{\mathcal{C}}^A$ (respectively Equation (min-norm)).*

In other words, for $N$ the number of non-empty activation cones, Theorem 1 states that at least $(1-\varepsilon)N$ cones satisfy items (i) and (ii) above. In particular, for a large overparametrization (having $m \gtrsim \min(n^d, 2^n)$) is here sufficient to have a favorable loss landscape, i.e., having only few regions with spurious local minima and most of them with global ones. Moreover, the notion of spurious local minimum in Theorem 1 is restricted to $\overline{\mathcal{C}}^A$, which is a stronger notion of local minimality[1] and thus leads to stronger favorability properties of the loss landscape. This large overparametrization requirement is actually necessary for favorability, as illustrated on the orthogonal example in Section 6.

In comparison, Karhadkar et al. (2024) showed that a mild overparametrization ($m \gtrsim n/d$) is sufficient to get only a small fraction of cones with spurious local minima for the unregularized problem. Although it remains unknown whether such a mild overparametrization also leads to global minima of the unregularized problem in most of the cones (i.e., the first point of Theorem 1), it still suggests, from a loss landscape point of view, that reaching a global minimum of the unregularized problem is generally much easier than the regularized one. While such an observation seems intuitive, Theorem 1 precisely quantifies this difference: while the unregularized landscape is *favorable* with $m \gtrsim n/d$ neurons, the regularized landscape only becomes favorable with $m \gtrsim \min(n^d, 2^n)$ neurons. Note that such a difference is not due to a possible looseness in our bound, since Theorem 3 below justifies that such a level of overparametrization is required to get a favorable regularized landscape.

This difference between regularized and unregularized landscapes may seem counterintuitive, since adding a regularization term $\lambda\|\theta\|^2$ is often expected to make the objective "more convex" and thus easier to minimize. However, in the absence of regularization, the set of global minima (i.e., interpolating solutions) is extremely large. Introducing even a small amount of regularization drastically reduces this set: while there are many ways to interpolate the data, only a few (if not unique; see, e.g., Boursier & Flammarion, 2023; Kamber & Parhi, 2026) achieve minimal norm. Consequently, the set of global minima becomes much more restricted with $\ell_2$-regularization, which in turn increases the level of overparametrization required for the loss landscape to be favorable.

Importantly, both Theorem 1 and Karhadkar et al. (2024) characterize the landscape in terms of the *number of cones*. However, in practice, cones can vary significantly in size, as some span larger angular regions of the weight space than others. Extending Theorem 1 to account for cone measures would require substantially more involved analysis and is therefore beyond the scope of this work. Nevertheless, a measure-based perspective is arguably more representative of practical settings, where weights are typically initialized from isotropic distributions such as Gaussians. In Appendix A, we complement our main experiments by considering Gaussian initialization, instead of sampling cones uniformly at random, to better reflect this viewpoint. Under this setting, we observe qualitatively similar behavior, although the landscape may appear more favorable under milder overparametrization for certain data structures (see Figures 3 to 7).

**Sketch of proof.** Thanks to the results of Wang et al. (2022), using a convex problem equivalent to the problem in Equation (Reg-$\lambda$), there exists a globally minimizing neural network with only $n+1$ non-zero neurons. From there, any cone containing (at least) $n+1$ neurons with the same activation pattern as the $n+1$ neurons of this optimal network can be shown to satisfy the two properties of Theorem 1 (see Lemma 1). We then show that when choosing an activation cone uniformly at random, there is a high probability to get such $n+1$ neurons. Actually, this is equivalent to the coupon collector problem (see, e.g., Feller, 1991): drawing $m$ independent coupons uniformly at random among a set of $\mathcal{O}\left(\min(n^d, 2^n)\right)$ coupons –

---

[1]Any local minimum in $\Theta$ is indeed also a local minimum in any subset $S \subseteq \Theta$ containing this point.

which are all the possible neuron activation patterns – and lower bounding the probability that $n+1$ *winning* coupons are collected within these $m$ random coupons allows to conclude. □

On its own, the loss landscape result of Theorem 1 justifies that when selecting an interpolator of the training data at random with large overparametrization, there is a high probability that the obtained estimator is small norm and even that it might be close to a global minimum of Equation (min-norm). This observation is directly related to the fact that selecting a neural network with weights sampled at random, conditioned on the fact that it interpolates the data, yields a good generalization to new unseen data (Chiang et al., 2023; Buzaglo et al., 2024). Indeed such a network sampled at random should be of small norm with high probability and should thus generalize well.

## 5 Connecting loss landscape with optimization dynamics

Although insightful and directly related to a random selection of the weights, loss landscape results such as Theorem 1 do not provide any guarantee on the convergence of typical optimization schemes towards global minima of the objective. In this section, we consider subgradient flow on the objective loss of Equation (Reg-$\lambda$) with a small regularization parameter $\lambda$, i.e., a solution of the following differential inclusion for almost any $t \in \mathbb{R}_+$:

$$\dot{\theta}(t) \in -\partial_\theta L_\lambda(\theta) \tag{3}$$
$$\text{where } L_\lambda(\theta) = \frac{1}{n}\sum_{k=1}^{n}(f_\theta(x_k) - y_k)^2 + \lambda\|\theta\|_2^2.$$

Due to the non-convexity of the objective function $L_\lambda$, the initialization is known to play a key role. In particular, large initializations are known to lead to the Neural Tangent Kernel (NTK) regime in the unregularized case. In this regime, the training resembles random features, where the inner layer is fixed at initialization and only output layer weights are adjusted during training (Chizat et al., 2019). In the case of small regularization, the dynamics are more complex but still follow the NTK regime at the beginning. It is only after reaching near interpolation of the training data that *grokking* is happening, where inner layer weights will also be adjusted until reaching a stationary point of $L_\lambda$ (Power et al., 2022; Liu et al., 2023; Lyu et al., 2024; Boursier et al., 2025). We believe that the loss landscape result of Theorem 1 is relevant in this large initialization regime, as the inner layer weights are nearly chosen at random at the beginning of the grokking phase. From here, the dynamics are driven by the regularization parameter (while maintaining interpolation) and should converge to a nearby local minimum. In particular, if most of the activation cones only include global minima, there is a good chance to converge to such a global minimum given the random nature of the training dynamics – that largely depends on the randomly selected initialization. Such an intuition is empirically confirmed in Section 7, although a deeper understanding of this grokking phase remains to be theoretically developed.

In opposition when considering small scale initializations, the features change drastically before reaching interpolation. They do so following the implicit bias of gradient flow on the unregularized loss, which annihilates the random features approximation once interpolation is reached. While this implicit bias for small initialization has been shown to coincide with the global minimum of Equation (min-norm) in multiple works, Chistikov et al. (2023) provided a family of data examples where the unregularized gradient flow instead converges to a local minimum of Equation (min-norm) that is not global, but benefits other regularities – i.e., a sparse representation. Building on those examples, we can show that even with regularization and arbitrarily large overparametrization, the parameters will converge towards an interpolating stationary point which is not a global minimum of Equation (Reg-$\lambda$) in the small initialization regime, but is instead a sparse interpolator of the data.

More precisely, for any $d \geq 3$, first we fix centers $(\widehat{x}_k)_{k \in [d]}$ as the following unit vectors:

$$\widehat{x}_1 := e_1, \qquad \widehat{x}_2 := \tfrac{8}{9}e_1 - \tfrac{4}{9}e_2 + \tfrac{1}{9}e_3, \qquad \widehat{x}_3 := \tfrac{8}{9}e_1 + \tfrac{4}{9}e_2 + \tfrac{1}{9}e_3$$
$$\widehat{x}_k := \tfrac{8}{9}e_1 + \tfrac{\sqrt{17}}{9}e_k \quad \text{for all } k \geq 4;$$

and we fix teacher $v_\star$ as the unit vector

$$v_\star := \tfrac{4}{5}e_1 + \tfrac{3}{5}e_3.$$

Then the next assumption details our requirements on the training dataset. To streamline the presentation, we assume that $n = d$, i.e., the number of samples equals the dimension, and that the points $x_k$ are unit vectors. In part (a), we assume that each point $x_k$ is near the corresponding center $\widehat{x}_k$, namely their cosine is at least $1 - \eta$ where $\eta > 0$ is a small threshold depending only on $d$, and also that each label $y_k$ is given by the inner product with the teacher $v_\star$. In part (b), we exclude some special cases of the empirical covariance matrix $H$ that do not decrease the Lebesgue measure.

**Assumption 1.** *We consider training datasets that consist of points $X := (x_k)_{k \in [d]} \in (\mathbb{S}^{d-1})^d \subseteq \mathbb{R}^{d \times d}$ and labels $(y_k)_{k \in [d]} \in \mathbb{R}^d$ such that:*

    *(a) for all $k \in [d]$, we have* $\qquad x_k^\top \widehat{x}_k > 1 - \eta \qquad$ *and* $\qquad y_k = v_\star^\top x_k$;

    *(b) the eigenvalues of $H := \frac{1}{d} X X^\top$ are distinct, and $v_\star$ is not in a span of fewer than $d$ eigenvectors of $H$.*

Note that the set of all training point matrices $X$ that we consider has non-zero Lebesgue measure in $(\mathbb{S}^{d-1})^d$, so it cannot be regarded as a single special case.

Below, our second assumption in this section concerns the subgradient flow in Equation (3). In part (a), we assume that the network is initialized randomly with scale $\alpha$ (uniformly in direction for the inner layer, and uniformly in sign for the output layer) where $\alpha > 0$ is small constant, and that the two layers are balanced. The balancedness at initialization is a standard assumption in the literature (see, e.g., Boursier et al., 2022; Chistikov et al., 2023; Min et al., 2024), and it would be not difficult although technically complicating to relax it to domination of the inner by the output layer, i.e., $\|w_i(0)\| \leq |a_i(0)|$ for all $i \in [m]$. In part (b), we exclude some unrealistic flows that might otherwise be possible theoretically due to the use of the subdifferential: we require that, whenever a neuron deactivates on all training points, then it stays deactivated for the remainder of the training. Note this assumption is representative of practice, since common implementations of backpropagation consider $\sigma'(0) = 0$ for the ReLU activation, and has been previously considered by Min et al. (2024, Definition 1).

**Assumption 2.** *For all $i \in [m]$ we have*

    *(a) $w_i(0) \overset{iid}{\sim} \mathcal{U}(\alpha \mathbb{S}_{d-1})$ and $a_i(0) \overset{iid}{\sim} \mathcal{U}(\{-\alpha, \alpha\})$;*

    *(b) for any $t \in \mathbb{R}_+$, if $w_i(t)^\top x_k \leq 0$ for all $k \in [d]$, then $w_i(t')^\top x_k \leq 0$ for all $t' \geq t$ and all $k \in [d]$.*

Let $\mu_{\min}$ denote the smallest eigenvalue of the empirical covariance matrix $H$.

**Theorem 2.** *Let Assumptions 1 and 2 hold for small enough $\eta$. With probability at least $1 - (\frac{3}{4})^m$, and for all $\varepsilon \in (0, \frac{1}{2}]$, there exists $\alpha^\star > 0$ such that, for all initialization scales $\alpha \leq \alpha^\star$ and all regularization parameters $\lambda \leq \mu_{\min} \alpha^\varepsilon$, every subgradient flow in Equation (3) converges to a network $\theta_\infty = \lim_{t \to \infty} \theta(t)$ such that:*

    *(i) the mean square error $\frac{1}{d} \sum_{k=1}^d (f_{\theta_\infty}(x_k) - y_k)^2$ is at most $\lambda^2 / \mu_{\min}$;*

    *(ii) for any $x \in \mathbb{R}^d$, $f_{\theta_\infty}(x) = \sigma(x^\top u_\star)$ where $u_\star$ is defined by Equation (7) in Appendix C;*

    *(iii) if $m \geq 2$ then $\theta_\infty$ is not a global minimum of the regularized loss $L_\lambda$.*

**Sketch of proof.** We first establish that, if $m \geq 2$ then for every dataset that satisfies Assumption 1 and small enough $\eta$ and $\lambda$, although the training labels are given by inner products with a single teacher vector, it is impossible to globally minimize the regularized loss $L_\lambda$ by a rank-1 network, i.e., where all neurons are non-negative scalings of a single neuron. In particular, we show how the ReLU non-linearity makes it possible to construct a rank-2 network for which $L_\lambda$ is smaller than the minimum over all rank-1 networks.

The remainder of the proof consists of a detailed analysis of the subgradient flow starting from a scale-$\alpha$ random initialization as in Assumption 2, which with probability at least $1 - (\frac{3}{4})^m$ has at least one neuron with positive output weight and active on at least one training point. We delineate and analyze three phases of the training as follows.

1. We show that, for small enough $\alpha$, the weight decay due to the $\ell_2$ regularization reinforces a first alignment phase in which every neuron remains small, and either aligns closely to a single direction determined by the training dataset, or deactivates from all training points.

2. Building on Chistikov et al. (2023), we show that the next phase, which is much more complex due to simultaneous growing and turning of the active neurons, proceeds sufficiently fast so that the effect of the weight decay is limited, and concludes with the active neurons still closely aligned and their composite vector being at most distance $\alpha^{\varepsilon/2}$ away from the teacher vector.

3. Finally we show that, in the last phase, the deactivated neurons converge to 0, whereas the rest converge to perfect alignment with their composite vector tending to the point that minimizes the regularized loss by trading off the mean square error with the $\lambda$-scaled $\ell_2$ norm. In the most involved part of the proof, inspired by Chatterjee (2022) but complicated by the prominent presence of weight decay in this phase, we provide a novel argument for a local Polyak-Łojasiewicz inequality, which then enables us to bound any disalignment of the neurons that may temporarily occur. □

Theorem 2 can therefore be seen as a new kind of testimony to the double-edged power of the early alignment phase in the small initialization regime. Namely, in unregularized regression settings, the early alignment was recently shown to be able to lead both to a failure of interpolation (Boursier & Flammarion, 2025a) and to enhanced generalization through minimization of population loss (Boursier & Flammarion, 2025b). In contrast, we have now demonstrated that it is also able to render ineffective $\ell_2$ regularization with arbitrarily small parameter $\lambda$, i.e., to prevent weight decay from steering the optimization towards interpolators with smaller norm if that would involve increasing their rank. This is in line with the recent insights of D'Angelo et al. (2024), indeed Theorem 2 shows that in its setting, weight decay does not lead to near interpolators of smaller norm but has other beneficial properties that promote model sparsity – the model being equivalent to a single neuron network. In this sense, Theorem 2 can be seen as a positive result: even when small initialization leads to convergence to a spurious local minimum, the obtained solution still possesses desirable simplicity properties such as sparsity of its representation.

We remark that the limit networks $\theta_\infty$ in Theorem 2 are local minima only if they do not contain any zero neurons (and the probability of that decays exponentially in network width $m$), otherwise they are saddle points of the regularized loss because such zero neurons can be used to compose small perturbations that maintain the mean square error but decrease the network norm. Thus, again in contrast to unregularized regression where early alignment can break interpolation by causing convergence to stationary points that are usually local minima (Boursier & Flammarion, 2025a), here early alignment can disarm weight decay but not completely as the latter nevertheless ensures that the limit is usually a saddle of $L_\lambda$.

In this section we showed that the favorability of the loss landscape established in Theorem 1 might not fully explain the observed optimization the small initialization regime, where sparse interpolators might be preferred. Indeed, Theorem 2 holds for all network widths $m \geq 2$ and in all dimensions $d \geq 3$. Along the way, we performed a detailed analysis of the optimization in the setting of Theorem 2.

## 6 Is mild overparametrization sufficient? Case of orthogonal data

Theorem 1 claims that, when $m \gtrsim \min(2^n, n^d)$, most of activation cones contain a global minimum, making the loss landscape favorable. Such a favorability appears for milder regimes ($m \gtrsim n/d$ is sufficient) without any regularization. A fundamental question is then: *how tight is the requirement on $m$ in Theorem 1? Is the loss landscape favorable for milder overparametrizations?*

This section answers this question by considering the example of orthogonal data inputs. On this example, Theorem 3 shows that the requirement $m \gtrsim \min(2^n, n^d)$ is indeed necessary for Theorem 1 without any further data assumption. Moreover, it also illustrates more generally that gradient flow converges to a spurious stationary point of Equation (Reg-$\lambda$) with high probability when $m \lesssim \min(2^n, n^d)$ for any typical initialization scheme (independently of its scale).

**Assumption 3.** *The data inputs are pairwise orthogonal: for any $j, k \in [n]$ such that $j \neq k$, $x_j^\top x_k = 0$.*

For Theorem 3 below, we need to define two data dependent quantities:

$$\lambda^\star = \min\left(\sqrt{\sum_{k,y_k>0} y_k^2\|x_k\|^2}, \sqrt{\sum_{k,y_k<0} y_k^2\|x_k\|^2}\right),$$

$$n^\star = \max\left(\#\{k\in[n] \mid y_k > 0\}, \#\{k\in[n] \mid y_k < 0\}\right).$$

**Theorem 3.** *Consider data satisfying Assumption 3, network width $m \geq 2$ and regularization parameter $0 < \lambda \leq \lambda^\star$. Then only a fraction at most $m2^{-n^\star}$ of activation cones $\mathcal{C}^A$ are such that their closure $\bar{\mathcal{C}}^A$ contains a global minimum of either Equation* (Reg-$\lambda$) *or Equation* (min-norm).

*Moreover, consider the gradient flow solution $\theta$ of Equation* (3). *If the inner layer weights $w_i(0)$ are initialized independently with a rotation invariant distribution, then with probability at least $1 - m2^{-n^\star}$: any limit point of $\theta(t)$ as $t\to\infty$ is not a global minimum of $L_\lambda$.*

Note that for typical data (e.g., if $y_k \neq 0$ for all $k$), $n^\star \geq \frac{n}{2}$. As a consequence, Theorem 3 implies that as long as the number of neurons $m$ is small with respect to $2^{\frac{n}{2}}$, only a small fraction of activation cones contain global minima of the regularized (or min-norm) loss in their closure. This suggests that the requirement $m \gtrsim \min(2^n, n^d)$ – note that in the orthogonal case $2^n \leq n^d$ – of Theorem 1 is somewhat necessary to have a favorable loss landscape.

While loss landscape claims are generally agnostic of the dynamics encountered during the training of the network, Theorem 3 also claims that this $m \gtrsim \min(2^n, n^d)$ condition is also required to guarantee, with high probability, convergence towards a global minimum of the regularized loss. In other words, Theorem 3 implies that as soon as $m \lesssim \min(2^n, n^d)$, the network will not converge towards the global minimum of the regularized loss $L_\lambda$ when trained via gradient flow, even with large initializations.

In the unregularized case with orthogonal data, Boursier et al. (2022) have shown that with $m \gtrsim 2^n$ neurons and a small initialization, gradient flow converges arbitrarily close to a global minimum of Equation (min-norm). Moreover, Dana et al. (2025) recently showed that with only $m \gtrsim \log(n)$ neurons, the unregularized flow still converges towards an interpolator of the training data, without any guarantee on the norm of the obtained estimator. Since the solutions of Equation (min-norm) and Equation (Reg-$\lambda$) correspond as $\lambda \to 0$, Theorem 3 completes the picture by showing that below this $m \gtrsim 2^n$ threshold, the convergence point is not a global minimum of Equation (min-norm), although it interpolates the data.

**Sketch of proof.** In the orthogonal data case, any hidden weight neuron $w_i(t)$ has a constant activation pattern over time (see Lemma 7 in the appendix), so that for the limit point to be a global minimum of Equation (Reg-$\lambda$) (or Equation (min-norm)), one needs the activation cone of hidden weights to include at initialization a global minimum of the problem. Moreover in the orthogonal case, a cone includes a global minimum if and only if it contains two precise activation patterns (see Lemma 6).

From there, we can again reduce the problem to a coupon collector problem: drawing $m$ coupons uniformly at random – given by the activation patterns of the hidden weights at initialization – among a set of $2^n$ possible patterns, we can then upper bound the probability to collect the two winning tickets described by Lemma 6. $\qquad\square$

## 7 Experiments

Section 7.1 presents experimental results supporting the favorable landscape properties established in Theorem 1, while Section 7.2 illustrates the training dynamics described in Section 5. Additional experiments related to Sections 5 and 6 are provided in Appendix A.

### 7.1 Loss landscape

Figure 1 illustrates the proportion of non-empty activation cones, whose closure contains a global minimum or local minimum of Equation (Reg-$\lambda$) with $\lambda = 0.01$, for different values of width $m$, number of data points $n$

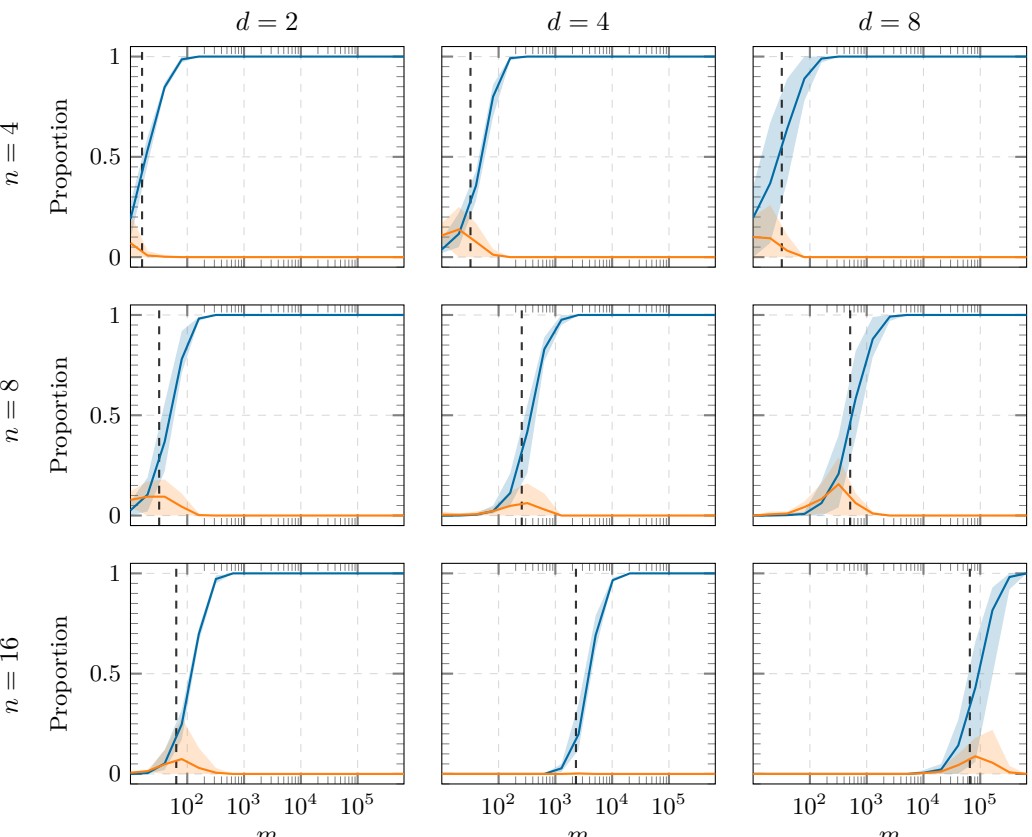

Figure 1: Proportion of activation cones containing global minima (blue) and spurious local minima (orange) across varying $m$, $n$, and $d$. The vertical dotted line corresponds to the number of non-empty neuron activation patterns, of which there are $4 \cdot \sum_{i=0}^{d-1} \binom{n-1}{i} = \mathcal{O}\big(\min(2^n, n^d)\big)$ many.

and data dimension $d$. Data is generated by drawing independent data points $x_i$ at random according to a standard Gaussian distribution, where the labels $y_i$ are given by the corresponding output of a teacher two-layer ReLU network of width 10. Shaded areas correspond to the min/max deviations observed over 5 random datasets. Experimental details can be found in Appendix A.

As predicted by Theorem 1, the fraction of non-empty activation cones containing global minima of Equation (Reg-$\lambda$) approaches 1 as soon as the network width exceeds the number of non-empty neuron activation patterns, equal to $4 \cdot \sum_{i=0}^{d-1} \binom{n-1}{i}$ thanks to Cover (1965), which is upper bounded by $\mathcal{O}\big(\min(2^n, n^d)\big)$. Before this overparametrization level, only a few cones contain global minima of Equation (Reg-$\lambda$), confirming the tightness of our bound in Theorem 1 and that it is not only specific to the orthogonal data case.

Additionally, the number of cones containing spurious local minima is close to 0 after this threshold. Maybe surprisingly, this fraction remains small for smaller values of the width – in contrast with the orthogonal case (see Figure 4) where this fraction is close to 1 for small $m$. However, it still reaches a significant value (of order 0.1) before the width threshold, suggesting that convergence towards a spurious local minimum is significantly probable for these values of the width.

## 7.2 Training dynamics

In this section, we consider the training setup introduced in Section 5, with input dimension $d = 5$. We train a ReLU network of width $m = 100$ using gradient descent with a small $\ell_2$-regularization parameter $\lambda = 10^{-5}$. Additional experimental details are provided in Appendix A.2.1.

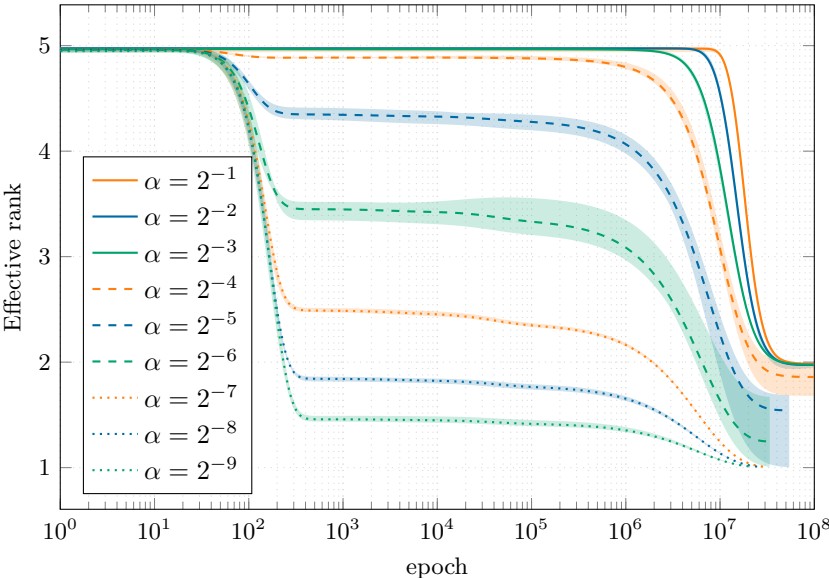

Figure 2: Evolution of effective rank during training for different initialization scales $\alpha$. We plot the mean over 5 different runs. The shaded areas correspond to the min/max deviations observed over those 5 runs. For each run, both the dataset and the initial weights are drawn from the distribution discussed in Appendix A.2.1.

Figure 2 shows the evolution of network features during training for different initialization scales $\alpha$. Specifically, it tracks the effective rank (defined as a continuous proxy for matrix rank; see, e.g., Roy & Vetterli, 2007) of the hidden weight matrix. The figure highlights the two regimes predicted in Section 5.

- For large initialization scale $\alpha$, the effective rank is initially high (equal to 5, corresponding to full rank) and remains so for a long period, even after interpolation is achieved (around $10^3$ epochs; see Figure 12 in Appendix A). This behavior is characteristic of the *lazy regime*, where feature representations remain dense throughout interpolation. A reduction in rank only occurs much later, during a *grokking* phase (around $10^7$ epochs), when the effective rank decreases toward that of the minimal-norm solution.

- For small initialization scale, the effective rank starts at the same value but rapidly decreases within fewer than $10^3$ epochs, indicating an early transition to a low-rank representation. This reflects the feature learning regime, and in particular the early alignment phenomenon described in the proof of Theorem 2, where weights quickly concentrate along a few dominant directions. The effective rank then remains low throughout training, with a further reduction during the later grokking phase.

A complementary perspective is provided by Figure 11 (see Appendix A), which tracks the number of neuron directions and leads to similar conclusions. Notably, while Theorem 2 and Figure 8 show that small initialization can result in larger parameter norms in this example, the effective rank (and number of neuron directions) is lower in this regime. This suggests that small initialization promotes a simpler representation, better captured by low-rank structure or the number of neuron directions than by parameter norm alone.

## 8    Conclusion

In this work, we quantified under what level of overparametrization the regularized loss landscape with two-layer ReLU networks is favorable. However, this favorability comes at the expense of a large overparametrization (in contrast with the unregularized case), and is not necessarily relevant to the training dynamics happening for small initializations.

In Theorem 2, the scale bound $\alpha^\star$ depends on the random directions of the hidden neurons and the random signs of the output weights at network initialization. Future work could seek to remove the latter dependencies and to bound the regularization parameter $\lambda$ independently of $\alpha$. Moreover, the final estimator still has

a simple structure, enforcing the idea that weight decay is not necessarily helpful to reach minimal norm interpolators, but might however lead to simple interpolators with good generalization properties (D'Angelo et al., 2024). Exploring the properties of such spurious local minima is a challenging and interesting direction, left open for future work.

Additionally, we showed that the overparametrization $m \gtrsim \min(2^n, n^d)$ is necessary to reach minimal norm interpolators. However, such a level of overparametrization is typically unrealistic and undesirable in practical settings due to its prohibitive computational costs. With fewer parameters however, it is not clear how far from *norm minimality* would the final estimator be. While this estimator might not be optimal, we believe it could still be relatively good (e.g., having a norm only slightly larger than the minimal one) and generalize well, which could explain why such a level of overparametrization might not be required in practice. Such a question is also left open for future work.

## 9   Acknowledgements

The authors acknowledge the Engineering and Physical Sciences Research Council (EPSRC) and DIMAP research centre at the University of Warwick for partial support. M. Bowditch is supported by the EPSRC through the Mathematics of Systems II Centre for Doctoral Training at the University of Warwick (reference EP/S022244/1). The authors acknowledge the use of the Batch Compute System in the Department of Computer Science at the University of Warwick, and associated support services, in the completion of this work. Additional computing facilities, also used for this work, were provided by the Scientific Computing Research Technology Platform of the University of Warwick.

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

# Appendix

## Table of contents

## A Additional experiments

### A.1 Experimental details

**Activation cones with optimal points.** To estimate the probabilities in Figure 1, we sample from the uniform distribution over the nonempty activation cones (i.e., sample a random element from the set $\{A \mid \mathcal{C}^A \neq \emptyset\}$). We then optimize the regularized loss Equation (Reg-$\lambda$) over $(W, a) \in \mathbb{R}^{m \times (d+1)}$ under the constraints that for all $i \in [m], k \in [n]$, $\mathbb{1}(w_i^\top x_k \geq 0) = A_{i,k}$ and $\mathbb{1}(a_i \geq 0) = A_{i,n+1}$, i.e., under the constraint that each neuron has the activation pattern that $A$ implies for that neuron.

To solve this convex objective under the given linear constraints, we first follow Wang et al. (2022) to remove duplicate rows of $A$, i.e., we keep at most one neuron for each activation pattern. This does not change the value of any local or global minimum that can be obtained (see Wang et al., 2022, Theorems 1 and 2). We then use Clarabel (Goulart & Chen, 2024) as a quadratic programming solver that is distributed with CVXPY (Diamond & Boyd, 2016) (Apache License 2.0) to solve the resulting optimization problem.

In addition, we find the globally optimal regularized loss, by solving the same problem, but with one neuron for every possible activation pattern. In other words, we take $\hat{m}$ large enough ($4 \cdot \sum_i^{d-1} \binom{n-1}{i}$) to be exact Cover, 1965) and choose $\hat{A} \in \{0, 1\}^{\hat{m} \times (n+1)}$ such that each row of $\hat{A}$ is distinct and $\mathcal{C}^{\hat{A}} \neq \emptyset$.

If the regularized loss obtained for $A$ is no larger than the globally optimal regularized loss plus the numerical tolerance $10^{-7}$, we consider $\bar{\mathcal{C}}^A$ to contain a global optimum.

We repeat this procedure 100 times for independently sampled $A$ and plot the proportion of nonempty activation cones that contain a global optimum in Figure 1 for $\lambda = 0.01$ and different values of $n$, $d$, and $m$. The plots show the min/max deviations over five different random datasets.

**Stationary points.** If an activation cone does not contain a global optimum, we check whether the point at which the regularized loss is minimized subject to the constraints implied by $A$, is a stationary point. Specifically, we compute the gradient of the weights $(W, a)$ at this point. For this, we define the derivative of the ReLU function at $x$ to be 0 for $x < -5 \cdot 10^{-5}$ and 1 for $x > 5 \cdot 10^{-5}$. For other values of $x$, we introduce tunable parameters, one for each ReLU, for the derivative that lies between 0 and 1. We then use the quadratic programming solver Clarabel to tune these parameters such that the norm of the gradient of the network is minimized. If the absolute values of the entries in the resulting gradient are all less than $5 \cdot 10^{-5}$, we declare the point a local minimum within the cone.

Figure 1 also shows the proportion of activation cones that have stationary points which are not optimal. Again this is based on the sample of 100 $A$s and over the five random datasets.

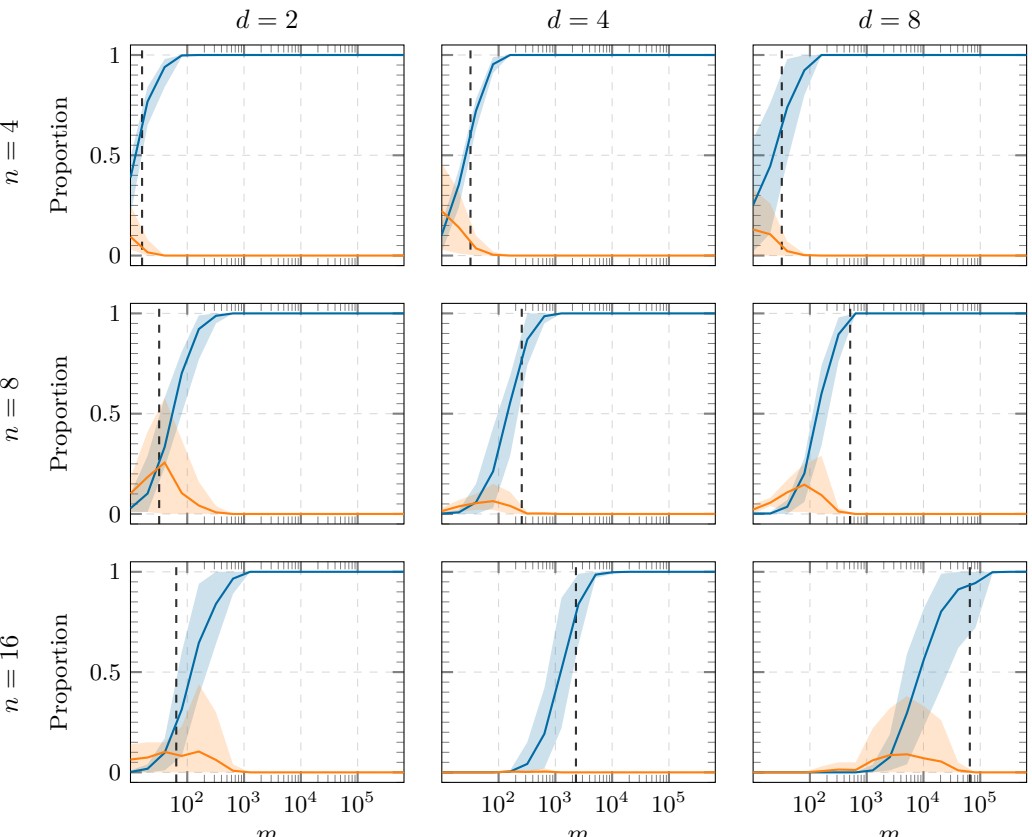

Figure 3: We sample a nonempty activation cone by generating a random network and observing the activation patterns that the neurons of the random network have. The plot shows the proportion of activation cones containing global minima (blue) and spurious local minima (orange) across varying $m$, $n$, and $d$, when sampled in this way. The vertical dotted line corresponds to the number of non-empty neuron activation patterns, of which there are $4 \cdot \sum_{i=0}^{d-1} \binom{n-1}{i} = \mathcal{O}\big(\min(2^n, n^d)\big)$ many.

**Alternative cone sampling procedures and datasets.** In the following we present further plots that show the impact of using a different procedure for sampling nonempty activation cones as well as results for the case that the data is orthogonal.

Instead of sampling nonempty activation cones uniformly at random, it may also be natural to instead take a random network of width $m$ and observe the implied activation cone of this random network. Concretely, we can sample a nonempty activation cone by generating $m$ random vectors $v_1, \ldots, v_m$ in $\mathbb{R}^d$ and $m$ scalars $a_1, \ldots, a_m$ with all numbers drawn independently from a standard Gaussian distribution. We then obtain the nonempty activation cone $A$ by defining for all $i \in [m]$ and $k \in [n]$, $A_{i,k} \coloneqq \mathbb{1}(w_i^\top x_k \geq 0)$ and $A_{i,n+1} \coloneqq \mathbb{1}(a_i \geq 0)$.

Figure 3 shows the same results as Figure 1 except that activation cones are sample in this alternative, non-uniform way. The observations are here similar to the ones of Figure 1. It confirms that Theorem 1, which considers the fraction of non-empty activation cones, is also relevant to typical neural network initializations, that draw the weights as i.i.d. Gaussian variables.

Figure 4 shows the same results as Figure 1 except that the data consists of $n$ random orthogonal unit vectors, with the labels again given by the corresponding output of a teacher two-layer ReLU network of width 10. Note that for such a dataset, the two different strategies for sampling nonempty activation cones (either

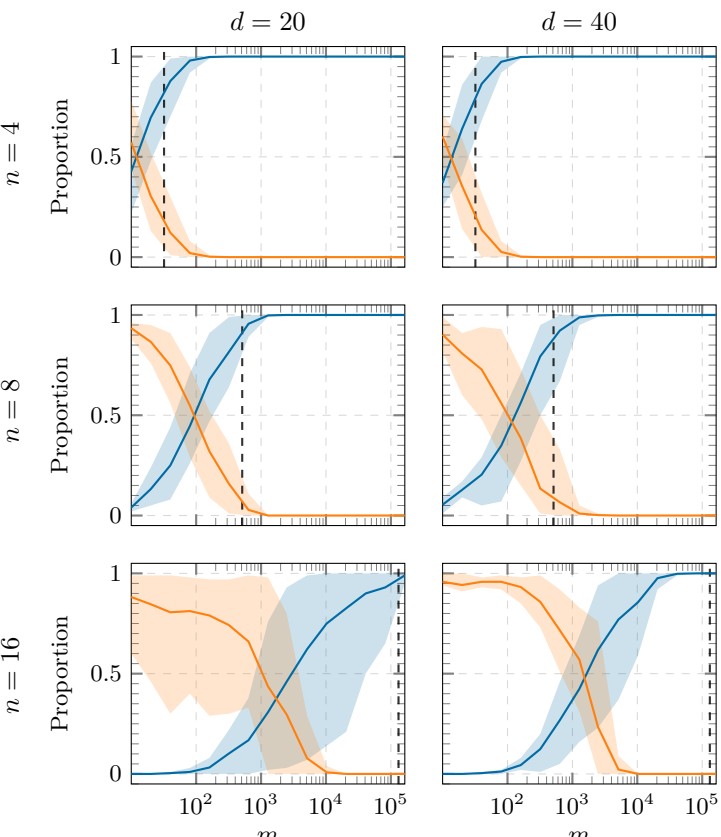

Figure 4: Proportion of activation cones containing global minima (blue) and local minima (orange) across varying $m$, $n$, and $d$ for orthogonal datasets. The vertical dotted line corresponds to the number of non-empty neuron activation patterns, of which there are $4 \cdot \sum_{i=0}^{d-1} \binom{n-1}{i} = \mathcal{O}\big(\min(2^n, n^d)\big)$ many.

uniformly at random or by observing the activation patterns of the neurons of a random network) both produce a uniform distribution over all nonempty activation cones.

As highlighted by Theorem 3, the orthogonal data case can be considered as worst case from a landscape point of view. Indeed, the transition to a favorable landscape also happens around the width threshold $4 \cdot \sum_{i=0}^{d-1} \binom{n-1}{i}$; but the fraction of cones with spurious local minima is here almost complementary to the one with global minima. In other words, (almost) every activation cone either contains a spurious local minimum or a global one. It makes the optimization harder, since the weights will converge towards a spurious local minimum of the regularized objective as soon as the number of parameters $m$ is small with respect to $2^n$, as predicted by Theorem 3.

Figure 5 shows the same results as Figure 1 except that the labels $y$ are chosen uniformly at random from $[-1, 1]$ instead of using a teacher network.

Figures 6 and 7 report the same experiment as Figure 1, except with a much larger number of data points $n$ or data dimension $d$, respectively.

In Figures 5 and 7, the behavior is consistent with that of Figure 1, the proportion of non-empty activation cones containing a global minima approaches 1 once the network width $m$ exceeds the number of non-empty neuron activation patterns. At the same time, the fraction of cones containing spurious local minima is close to 0 beyond this threshold.

Figure 6 however slightly differs from Figure 1, as the fraction of spurious local minima, even in the underparametrized regime, seems to vanish to 0 as $n$ grows large. We believe this is due to the specific data structure considered in our experiments, where labels are $y_i$ are given by the output of a small teacher

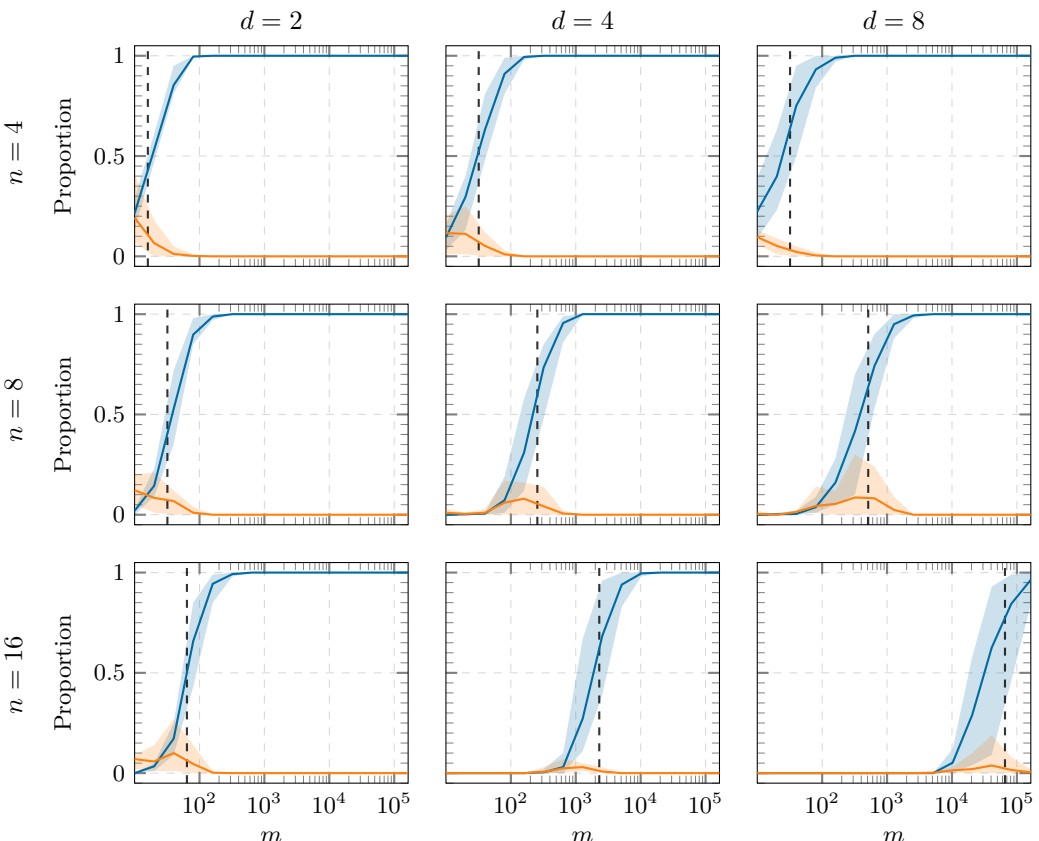

Figure 5: Labels $y$ are chosen uniformly at random from $[-1, 1]$. Shown are the proportions of activation cones containing global minima (blue) and spurious local minima (orange) across varying $m$, $n$, and $d$. The vertical dotted line corresponds to the number of non-empty neuron activation patterns, of which there are $4 \cdot \sum_{i=0}^{d-1} \binom{n-1}{i} = \mathcal{O}\big(\min(2^n, n^d)\big)$ many.

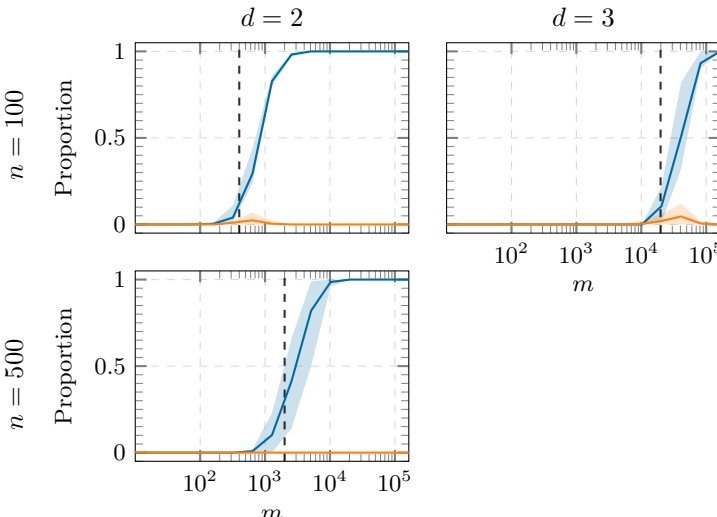

Figure 6: Proportion of activation cones containing global minima (blue) and bad local minima (orange) across varying $m$, $n$, and $d$, now for much larger $n$. The vertical dotted line corresponds to the number of non-empty neuron activation patterns, of which there are $4 \cdot \sum_{i=0}^{d-1} \binom{n-1}{i} = \mathcal{O}\big(\min(2^n, n^d)\big)$.

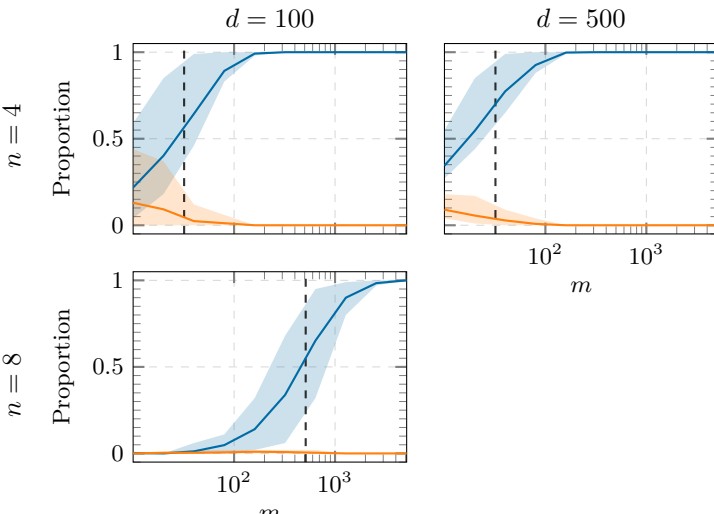

Figure 7: Proportion of activation cones containing global minima (blue) and bad local minima (orange) across varying $m$, $n$, and $d$, now for much larger $d$. The vertical dotted line corresponds to the number of non-empty neuron activation patterns, of which there are $4 \cdot \sum_{i=0}^{d-1} \binom{n-1}{i} = \mathcal{O}\big(\min(2^n, n^d)\big)$.

network and that in such a case (i.e., without any label noise), the *favorability threshold* should be much smaller (and actually correspond approximately to the number of parameters of the teacher network).

**Compute.** Experiments for Figures 1 and 3 take less than 10 CPU hours on Dual Intel Xeon E5-2643 v3 CPUs each. Experiments for Figure 4 take around 100 CPU hours on the same hardware.

### A.2 Experiments complementing Theorem 2

We investigate the impact of the initialization scale $\alpha$ on the properties of the minimum that gradient descent converges towards. The regularization parameter $\lambda$ should be sufficiently small such that we obtain near interpolator. Specifically, we choose $\lambda = 10^{-5}$.

Using data consistent with the setup of Theorem 2 (see details below), and a learning rate of 0.01, we find that the mean square error in all our experiments is in the region of $10^{-8}$. In other words, we obtain approximate interpolators.

However, Figure 8 shows the interpolator size (measured in the square of the Eucliden norm of all weights) and this size is markedly smaller for large initialization scales $\alpha$. As $\alpha$ decreases, there is a sharp transition and the size of the network becomes close to 2, which is the size that a network consisting only of the teacher neuron $v_\star$ would have.

Indeed, as Figure 9 shows, small initialization scales lead to convergence to a spurious local minimum of the regularized loss, whereas larger scales allow us to converge towards the global minimum. While near interpolation is always achieved, Figure 10 shows that small initialization also leads to a slightly worse unregularized loss.

The globally optimal solution for our chosen $\lambda$ and our generated data always consists of three distinct neurons, i.e., three neurons with different directions (and different activation patterns). Figure 11 shows, for $d = 5$ and different values of $\alpha$, how many different neuron directions the networks have during training. (More precise details can be found below.) We can see that after a few million epochs, the neurons start to align, reducing the number of distinct directions.

For small $\alpha$, this happens earlier in the process and we converge towards a solution consisting only of a single neuron direction, with a single activation pattern. Once we reach such a state in which all neurons are aligned

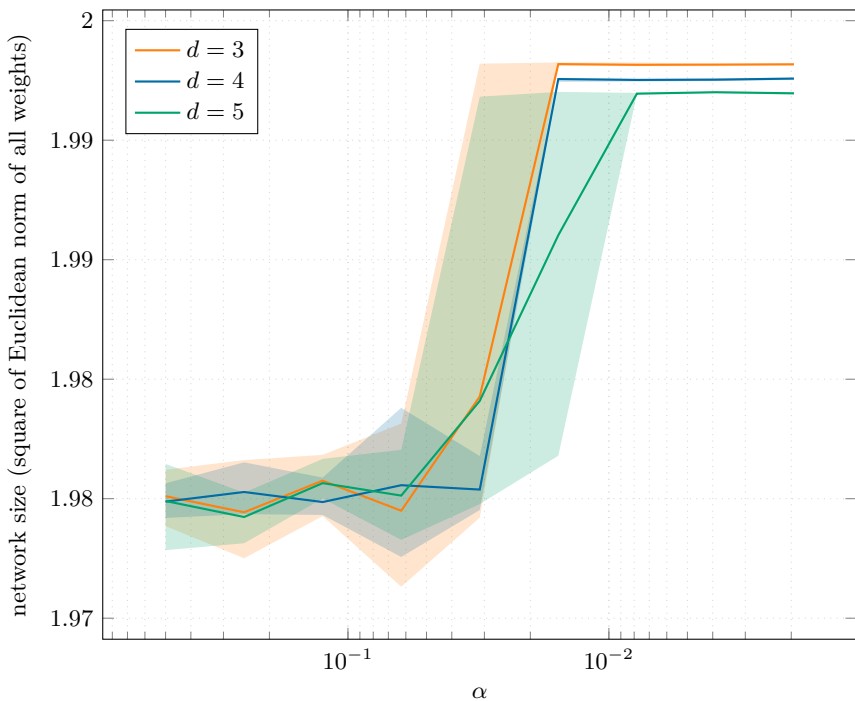

Figure 8: The square of the Euclidean norm of all network weights after training has finished, starting with different initialization scales $\alpha$ for $d$ dimensional data. The shaded areas correspond to the min/max deviations observed over 5 different runs. For each run, both the dataset and the initial weights are drawn from the distribution discussed in Appendix A.2.1. A network only consisting of the teacher neuron has a squared Euclidean norm of 2. We see that for large initialization scales, we converge to a network of notably smaller size.

in a single direction, the activation patterns of the aligned neurons never change again in our experiments, as this corresponds to a local minimum of the regularized loss.

For large $\alpha$ on the other hand, the alignment and reduction in distinct neuron direction happens later and we converge towards solutions involving three distinct neuron directions with three different activation patterns. These activation patterns match the activation patterns of the neurons in the optimal solution, i.e., they correspond to a global minimum of the loss.

### A.2.1 Experimental details

We generate data consistent with the setup of Theorem 2. Specifically, for $d \in \{3, 4, 5\}$, we define the centers $\hat{x}_i$ as in Section 5. For $i \in [d]$, to generate a data point $x_i$, we add Gaussian noise with zero mean and standard deviation 0.001 to the corresponding center and normalize the result to obtain a unit vector. Labels $y_i$ are given by the inner product of $x_i$ with the teacher $v_\star$.

We initialize a two-layer ReLU network of width 100 in a balanced way (again according to the setup in Section 5): first we initialize the first layer weights with independently drawn centered Gaussians with standard deviation $\alpha$, and then we initialize the second layer weight for each neuron by the norm of the first layer neuron with a random sign.

We train the network on the regularized loss with $\lambda = 10^{-5}$ and learning rate 0.01. We train until the square of the Euclidean norm of the gradient falls below $10^{-16}$ or until 100 million epochs, whichever occurs first.

Figure 8 shows the size of the resulting network (the square of the Euclidean norm of all weights) at the end of training for different values of $d$ and $\alpha$.

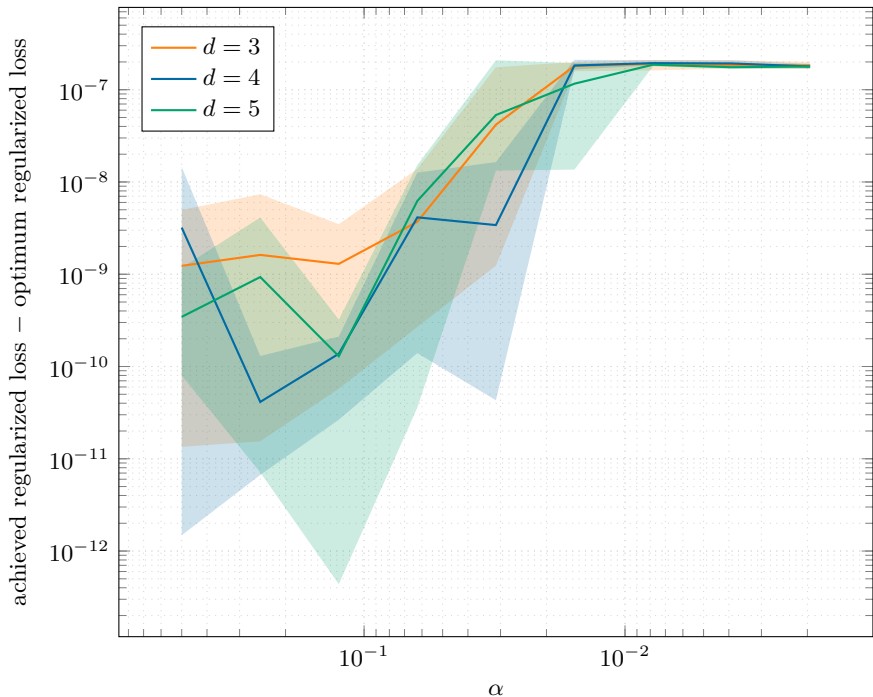

Figure 9: The difference between the final regularized loss of the trained network and the optimal regularized loss, for different initialization scales $\alpha$ for $d$ dimensional data. The shaded areas correspond to the min/max deviations observed over 5 different runs. For each run, both the dataset and the initial weights are drawn from the distribution discussed in Appendix A.2.1.

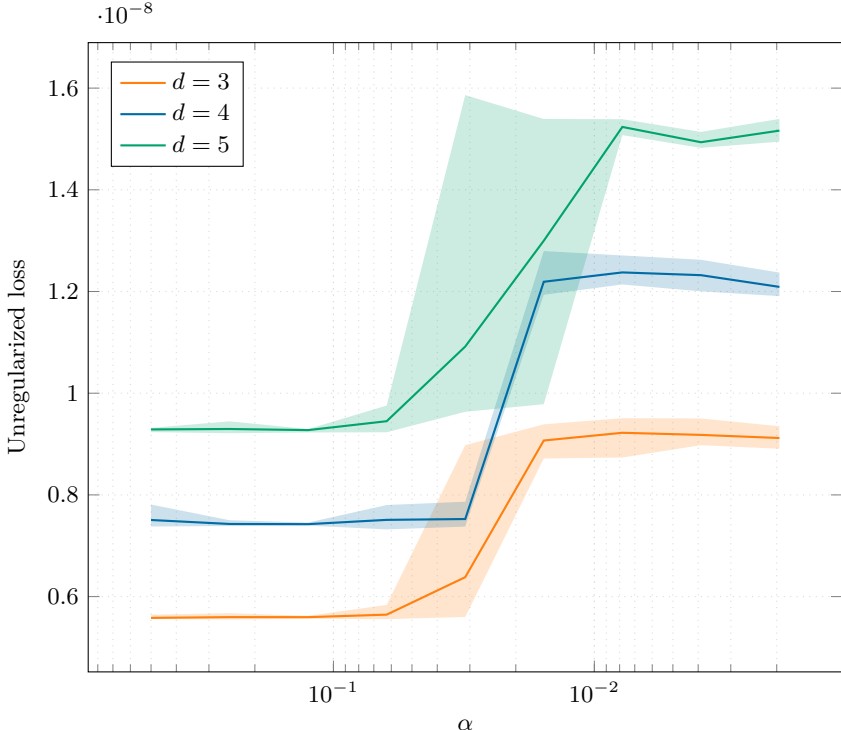

Figure 10: The final unregularized loss of the trained network, for different initialization scales $\alpha$ for $d$ dimensional data. The shaded areas correspond to the min/max deviations observed over 5 different runs. For each run, both the dataset and the initial weights are drawn from the distribution discussed in Appendix A.2.1.

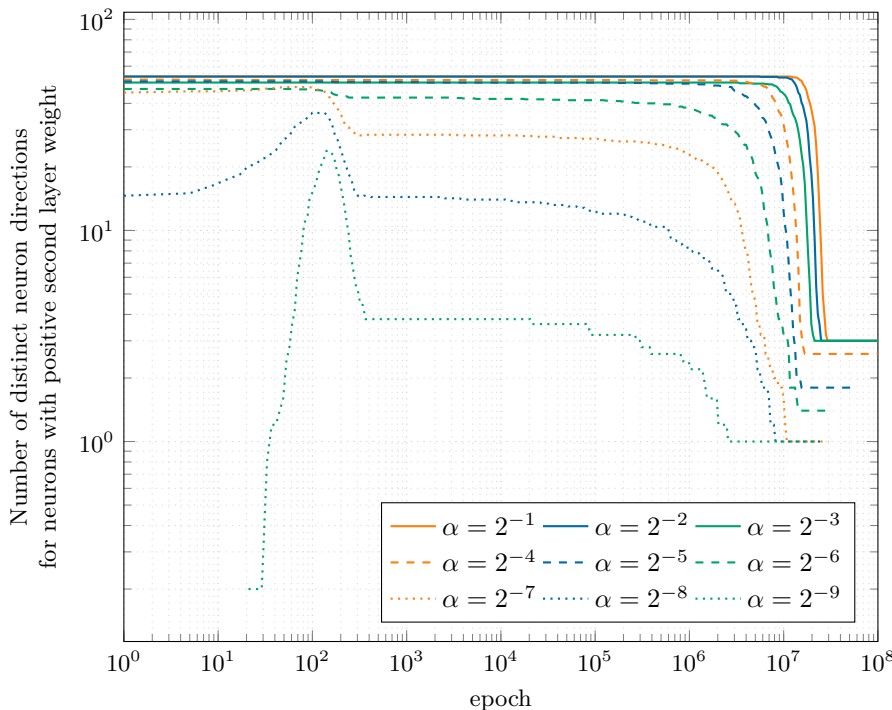

Figure 11: This figure shows the evolution of the network during training for different initialization scales $\alpha$ and $d = 5$. We plot the mean number of distinct neuron directions for neurons with positive second layer weight (as specified in Appendix A.2.1), where the mean is taken over 5 different runs. For each run, both the dataset and the initial weights are drawn from the distribution discussed in Appendix A.2.1. Shaded areas corresponding to min/max deviations are omitted for readability. Neurons with a norm below $10^{-4}$ are not counted, hence, for small initialization scales, initially there are no neurons to consider. Neuron norms then quickly increase past the $10^{-4}$ threshold before neurons start to align and the number of distinct neuron directions decreases until it reaches 1.

We also compute an optimal solution by solving the convex program as described in Appendix A.1. Figure 9 shows the difference between this optimal regularized loss and the regularized loss of the trained network.

For Figure 11, we determine the number of distinct neuron directions as follows: first we identify those neurons that have a positive second layer weight. We note that in our experiments, due to our specific data, the neurons with negative second layer weight always converge to 0 and are therefore not interesting to plot. We ignore neurons which, when scaled by their second layer weight, have norm less than $10^{-4}$. We then consider two neurons to have the same direction if they are at an angle of less than 0.1 radians.

All experiments were repeated five times and the plots show the mean values. Figures 2, 8 to 10 and 12 also show the min/max deviations. For Figure 11 the min/max deviations are omitted for readability, but after a large number of epochs, the number of neurons was always the same for all runs. The only exception are intermediate ranges for $\alpha$. Concretely, for $\alpha = 2^{-4}$, two runs resulted in two neuron direction, whereas the remaining three runs had three neuron directions. For $\alpha = 2^{-5}$, four runs resulted in two neuron directions, whereas the remaining run had only one neuron direction. And for $\alpha = 2^{-6}$, two runs resulted in two neuron directions and the remaining three runs in one neuron direction.

**Compute.** Experiments complementing Theorem 2 took around 35 CPU hours on Dual Intel Xeon E5-2643 v3 CPUs.

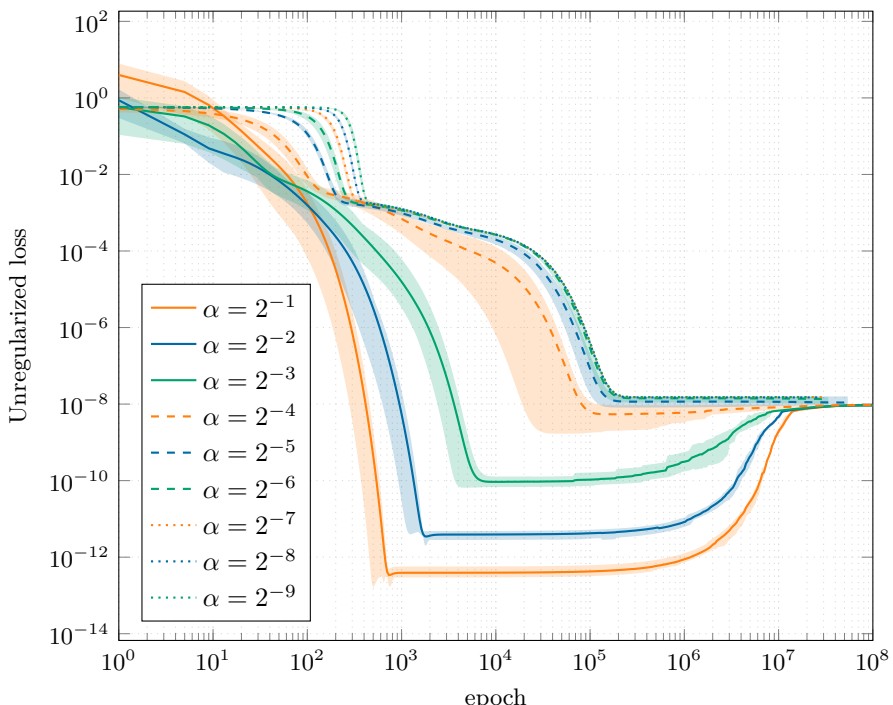

Figure 12: This figure shows the evolution of the unregularized loss during trained network, for different initialization scales $\alpha$ for $d = 5$ dimensional data. The shaded areas correspond to the min/max deviations observed over 5 different runs. For each run, both the dataset and the initial weights are drawn from the distribution discussed in Appendix A.2.1.

## B   Proof of Theorem 1

In this section, we will also use the notion of *neuron cone*, which is directly related to the activation cones defined in Equation (2). For that, we define the activation function

$$A_n : \begin{array}{l} \mathbb{R}^{d+1} \to \{0,1\}^{n+1} \\ (w,a) \mapsto (\mathbb{1}(w^\top x_k \geq 0)_{k \in [n]}, \mathbb{1}(a \geq 0)) \end{array} \, .$$

For any binary vector $u \in \{0,1\}^{n+1}$, we associate the neuron cone $\mathcal{C}_u \subseteq \mathbb{R}^{d+1}$ defined as

$$\mathcal{C}_u = A_n^{-1}(u).$$

Notably, for any binary matrix $A \in \{0,1\}^{m \times (n+1)}$ neuron cones and activation cones are related by the following equality:

$$\mathcal{C}^A = \prod_{i=1}^m \mathcal{C}_{A_i}.$$

In words, the parameters $(W, a) \in \mathbb{R}^{m \times (d+1)}$ belong to the activation cone $\mathcal{C}^A$ if and only if for any $i \in [m]$, the $i$-th neuron belongs to the neuron cone associated to the $i$-th row of $A$, i.e., $A_n(w_i, a_i) = A_i$.

Taking $m \geq n + 1$, Wang et al. (2022, Theorem 1) state that there exists a network with width $n + 1$ reaching the global minimum of Equation (Reg-$\lambda$). Similarly, Ergen & Pilanci (2021) show that there exists a network with width $n + 1$ reaching the global minimum of Equation (min-norm). From now, we only focus on Equation (Reg-$\lambda$), but our arguments can be directly extended to considering Equation (min-norm).

In other words, there exists $(W^\star, a^\star) \in \mathbb{R}^{m \times (d+1)}$ such that

$$L_\lambda(W^\star, a^\star) = \min_\theta \frac{1}{n} \sum_{k=1}^n (f_\theta(x_k) - y_k)^2 + \lambda \|\theta\|_2^2,$$

$$A_n(w_i^\star, a_i^\star) \neq A_n(w_j^\star, a_j^\star) \quad \text{for any } i, j \leq n+1 \text{ such that } i \neq j \tag{4}$$
$$\text{and} \quad w_i^\star = \mathbf{0}, a_i^\star = 0 \quad \text{for any } i > n+1.$$

Equation (4) states the additional condition that the global minimum of Equation (Reg-$\lambda$) has at most one (non-zero) neuron per neuron cone. Indeed, in the presence of several neurons inside a single cone, one can simply merge these neurons into a single one, which does not change the network output, while decreasing its norm (see Wang et al., 2022, Proposition 2).

From there using the permutation invariance of the parametrization, we can show that every activation cone containing at least one neuron $(w, a)$ such that $A_n(w) = A_n(w_i^\star)$ and $\text{sign}(a) = \text{sign}(a_i^\star)$ for all $i \in [n+1]$ necessarily contains a global minimum of the problem (and no spurious local minimum). This is stated formally by Lemma 1 below.

**Lemma 1.** *For any activation cone $\mathcal{C}^A$, if for any $i \in [n+1]$, there exists $j \in [m]$ such that $A_j = A_n(w_i^\star, a_i^\star)$, then:*

    *(i) $\bar{\mathcal{C}}^A$ contains a global minimum of Equation (Reg-$\lambda$);*

    *(ii) $\bar{\mathcal{C}}^A$ does not contain any spurious local minimum of Equation (Reg-$\lambda$) restricted to $\bar{\mathcal{C}}^A$.*

From Lemma 1, it is sufficient to count the fraction of non-empty cones $\mathcal{C}^A$ satisfying the following property

$$\text{for any } i \in [n+1], \text{ there exists } j \in [m] \text{ such that } A_j = A_n(w_i^\star, a_i^\star). \tag{5}$$

This can be done using a simple bound for the coupon collector problem.

**Lemma 2** (Lemma 11 by Karhadkar et al. 2024). *Let $\varepsilon \in (0, 1)$ and $p \leq q$ be positive integers. Let $Z_1, \ldots, Z_r$ be $r$ independent, uniformly at random variables in $[q]$. If $r \geq q \ln(\frac{p}{\varepsilon})$, then $[p] \subseteq \{Z_1, \ldots, Z_r\}$ with probability at least $1 - \varepsilon$.*

Indeed, note that if we choose uniformly at random a (non-empty) activation cone $\mathcal{C}^A$, it is equivalent to choosing independently, $m$ (non-empty) neuron cones $(\mathcal{C}_{u_i})_{i \in [m]}$ uniformly at random. Indeed, each $u_i \in \{0, 1\}^{n+1}$ then corresponds to the $i$-th row of the matrix $A \in \{0, 1\}^{m \times (n+1)}$. Thus, we have the following equality, assuming $A$ is a binary matrix in $\{0, 1\}^{m \times (n+1)}$, drawn uniformly at random among the set of matrices such that $\mathcal{C}^A$ is non-empty; and that the binary vectors $u_i$ are drawn i.i.d., uniformly at random among the set of binary vectors $u$ such that $\mathcal{C}_u$ is non-empty.

$$\mathbb{P}(A \text{ satisfies Equation (5)}) = \mathbb{P}(\forall i \in [n+1], \exists j \in [m], u_j = A_n(w_i^\star))$$
$$= \mathbb{P}\left(\{A_n(w_1^\star), \ldots, A_n(w_{n+1}^\star)\} \subseteq \{u_1, \ldots, u_m\}\right).$$

Thanks to Lemma 2, this yields that when $m \geq q \ln(\frac{n+1}{\varepsilon})$, $\mathbb{P}(A \text{ satisfies Equation (5)}) \geq 1 - \varepsilon$, where $q$ is the total number of non-empty neuron cones.

Moreover, Cover (1965) bounds the total number of non-empty cones as $q = \mathcal{O}(\min(2^n, n^d))$, which finally yields Theorem 1.

### B.1 Proof of Lemma 1

Lemma 1 is shown by means of merging, scaling and permuting the neurons, which are tools used in a long line of work (see, e.g., Ergen & Pilanci, 2021; Wang et al., 2022).

*Proof.* Consider $A \in \{0, 1\}^{m \times (n+1)}$ such that for any $i \in [n+1]$, there exists $j \in [m]$ such that $A_j = A_n(w_i^\star, a_i^\star)$.

As the cones of non-zero $w_i^\star$ are pairwise distinct, we actually have a permutation $\rho : [n+1] \to [n+1]$ such that for any $i \in [n+1]$: $A_{\rho(i)} = A_n(w_i^\star, a_i^\star)$. From there, simply note that the closure of the cone $\bar{\mathcal{C}}^A$ contains the zero matrix, and more generally zero neurons on any row of our choice. Notably, this implies that $(W^\star, a^\star)$, up to a permutation, belongs to $\bar{\mathcal{C}}^A$, i.e., $(W^{\star,\rho}, a^{\star,\rho}) \in \bar{\mathcal{C}}^A$ where

$$w^{\star,\rho}(\rho(i)) = w_i^\star,$$
$$a^{\star,\rho}(\rho(i)) = a_i^\star \quad \text{for any } i \in [m].$$

Since the objective function $L_\lambda$ is invariant under permutation, we then have $L_\lambda(W^\star, a^\star) = L_\lambda(W^{\star,\rho}, a^{\star,\rho})$. This proves the first point, i.e., that $\bar{\mathcal{C}}^A$ contains a global minimum of Equation (Reg-$\lambda$).

For the second point, consider a local minimum $(W, a) \in \bar{\mathcal{C}}^A$. Additionally, we assume in the following without loss of generality that $(W^\star, a^\star) \in \bar{\mathcal{C}}^A$ – i.e., we consider that the permutation $\rho$ described above is the identity. Also, we write

$$\beta_i = a_i w_i \quad \text{and} \quad \beta_i^\star = a_i^\star w_i^\star \quad \text{for any } i \in [m].$$

Note that rescaling any neuron $(w_i, a_i)$ to $(cw_i, \frac{1}{c}a_i)$ does not change the output function $f_\theta$, but changes the squared norm of the parameters $(W, a)$. As a consequence, it is known that any local minimum of Equations (Reg-$\lambda$) and (min-norm) is balanced, i.e., $\|w_i\|_2 = |a_i|$ for any $i \in [m]$ (Neyshabur et al., 2015; Savarese et al., 2019; Parhi & Nowak, 2021; Boursier & Flammarion, 2023). As a consequence, $(W, a)$ and $(W^\star, a^\star)$ are both balanced. Moreover, a direct consequence of this balanced property is the following equality

$$\|(W, a)\|_2^2 = 2 \sum_{i=1}^{m} \|\beta_i\|_2,$$

$$\|(W^\star, a^\star)\|_2^2 = 2 \sum_{i=1}^{m} \|\beta_i^\star\|_2.$$

From there, we can define for any $B \in \mathbb{R}^{m \times d}$ the alternative loss function as

$$\tilde{\mathcal{L}}(B) = \frac{1}{n} \sum_{k=1}^{n} (h_B(x_k) - y_k)^2 + 2\lambda \|B\|_{1,2},$$

$$\text{where } h_B(x_k) = \sum_{i=1}^{m} A_{ik} B_i^\top x_k$$

$$\text{and } \|B\|_{2,1} = \sum_{i=1}^{m} \|B_i\|_2.$$

For $\mathcal{B} \in \mathbb{R}^{m \times d}$ (respectively $\mathcal{B}^\star$) defined as the matrix whose rows are given by $\beta_i \in \mathbb{R}^d$ (respectively $\beta_i^\star \in \mathbb{R}^d$), note that

$$\tilde{\mathcal{L}}(\mathcal{B}) = \frac{1}{n} \sum_{k=1}^{n} (f_{(W,a)}(x_k) - y_k)^2 + \lambda \|(W, a)\|_2^2$$

$$\text{and } \tilde{\mathcal{L}}(\mathcal{B}^\star) = \frac{1}{n} \sum_{k=1}^{n} (f_{(W^\star, a^\star)}(x_k) - y_k)^2 + \lambda \|(W^\star, a^\star)\|_2^2.$$

Notably, note that the parametrization $h_B$ is linear in $B$. As a consequence, the function $\tilde{\mathcal{L}}$ is convex and we can thus define $\mathcal{B}^t = t\mathcal{B}^\star + (1 - t)\mathcal{B}$ for any $t \in [0, 1]$ such that :

$$\tilde{\mathcal{L}}(\mathcal{B}^t) \le t\tilde{\mathcal{L}}(\mathcal{B}^\star) + (1 - t)\tilde{\mathcal{L}}(\mathcal{B}).$$

Assume now that $(W, a)$ is a spurious local minimum, i.e., it is not a global one. In particular, the above inequality implies that for any $t \in (0, 1]$:

$$\tilde{\mathcal{L}}(\mathcal{B}^t) < \tilde{\mathcal{L}}(\mathcal{B}). \tag{6}$$

For any $t \in [0, 1]$, we can then define $(W^t, a^t) \in \mathbb{R}^{m \times (d+1)}$ as

$$a_i^t = \text{sign}(a_i) \sqrt{\|\beta_i^t\|_2},$$

$$W_i^t = \frac{\beta_i^t}{a_i^t},$$

where we omitted that we define $W_i^t = \mathbf{0}$ if $a_i^t = 0$. By convexity of the cone $\bar{\mathcal{C}}^A$, $(W^t, a^t) \in \bar{\mathcal{C}}^A$. Moreover, a quick computation directly yields that

$$\tilde{\mathcal{L}}(\mathcal{B}^t) = \frac{1}{n} \sum_{k=1}^{n} (f_{(W^t, a^t)}(x_k) - y_k)^2 + \lambda \|(W^t, a^t)\|_2^2.$$

It is also easy to check that $(W^t, a^t) \to (W, a)$ as $t \to 0$. Thanks to Equation (6), this then implies that $(W, a)$ is not a local minimum. By contradiction, it is a global minimum, which concludes the proof of Lemma 1. $\square$

## C  Proof of Theorem 2

In this proof, we write $\|v\|$ for the Euclidean norm of a vector $v$, $\|v\|_H := \sqrt{v^\top H v}$ for the energy norm with respect to a symmetric positive definite matrix $H$, $\bar{v} := v/\|v\|$ for the normalization of a non-zero vector $v$, and $\angle(u, v) := \arccos(\bar{u}^\top \bar{v})$ for the angle between non-zero vectors $u$ and $v$.

We shall also find it convenient to use $\epsilon := \varepsilon/2$, where $\varepsilon$ is as in the statement of Theorem 2.

We start by letting $(x_k^\dagger)_{k \in [d]}$ denote the rows of the inverse of the data matrix, i.e., the columns of $(X^{-1})^\top$, and with the following technical proposition.

**Proposition 1.** *Provided $\eta$ is sufficiently small, we have:*

   *(a) $(x_k)_{k \in [d]}$ are linearly independent;*

   *(b) $\angle(v_\star, x_k) < \pi/4$ for all $k \in [d]$;*

   *(c) $\cos \angle(v_\star, x_2^\dagger) > \sin \angle(x_2^\dagger, x_3^\dagger)$.*

*Proof.* Part (a) is straightforward to check.

It suffices to show (b) for $(\widehat{x}_k)_{k \in [d]}$, and to show (c) for the columns $(\widehat{x}_k^\dagger)_{k \in [d]}$ of $(\widehat{X}^{-1})^\top$, where $\widehat{X}$ is the matrix whose columns are $(\widehat{x}_k)_{k \in [d]}$.

For (b), for all $k \in [d]$ we have $v_\star^\top \widehat{x}_k \geq \frac{8}{9} \frac{4}{5} = \frac{32}{45} > \frac{1}{\sqrt{2}} = \cos(\pi/4)$.

For (c), first observe that:

$$\widehat{x}_2^\dagger = -\frac{9}{8} e_2 + \frac{9}{2} e_3 \qquad\qquad \widehat{x}_3^\dagger = \frac{9}{8} e_2 + \frac{9}{2} e_3.$$

Now we have

$$\cos \angle(v_\star, \widehat{x}_2^\dagger) = \frac{\frac{27}{10}}{\sqrt{\frac{81}{4} + \frac{81}{64}}} = \frac{12}{5\sqrt{17}}$$

and

$$\sin \angle(\widehat{x}_2^\dagger, \widehat{x}_3^\dagger) = \sqrt{1 - \left(\frac{\frac{81}{4} - \frac{81}{64}}{\frac{81}{4} + \frac{81}{64}}\right)^2} = \sqrt{1 - \frac{15^2}{17^2}} = \frac{8}{17},$$

so indeed the former is greater than the latter. $\square$

Note that, by Proposition 3 (a), we have $\mu_{\min} > 0$, so the empirical covariance matrix $H$ is positive definite.

We define $u_\star \in \mathbb{R}^d$ uniquely by

$$\bar{u}_\star^\top \overline{H(v_\star - u_\star)} = 1 \text{ and } \|H(v_\star - u_\star)\| = \lambda. \tag{7}$$

Then let $\Theta_{u_\star}$ denote the set of all network parameters $\theta = (W, a) \in \mathbb{R}^{m \times (d+1)}$ such that the two layers are balanced, the output weights are non-negative, the hidden neurons are non-negative scalings of $u_\star \in \mathbb{R}^d$, and the squares of the output weights sum to $\|u_\star\|$, i.e.,

$$a_i = \|w_i\| \quad \text{for all } i \in [m]$$

$$w_i = a_i\,\overline{u}_\star \quad \text{for all } i \in [m]$$
$$\textstyle\sum_{i\in[m]} a_i^2 = \|u_\star\|.$$

The next proposition consists of two parts. In (a), we establish that all networks in $\Theta_{u_\star}$ have the same value of the regularized loss $L_\lambda$, which we denote by $L_\lambda^\star$. Part (b) then states that, provided the network width is at least 2, the latter value is not minimal, and thus none of the networks in $\Theta_{u_\star}$ are global minima. In Lemma 5 below, we shall prove that $\lim_{t\to\infty}\theta(t)$ is a network in $\Theta_{u_\star}$, and the proof will also show that the latter are exactly the rank-1 minimizers of $L_\lambda$.

**Proposition 2.** *Provided $\eta$ and $\lambda$ are sufficiently small:*

*(a) for every $\theta \in \Theta_{u_\star}$ we have $L_\lambda(\theta) = \|v_\star - u_\star\|_H^2 + 2\lambda\|u_\star\| =: L_\lambda^\star$;*

*(b) if $m \geq 2$ then $\min\{L_\lambda(\theta) \mid \theta \in \mathbb{R}^{m\times(d+1)}\} < L_\lambda^\star$.*

*Proof.* Part (a) follows by observing that

$$\forall k \in [d], f_\theta(x_k) = u_\star^\top x_k =: z_k.$$

For (b), letting

$$\zeta := \cos \measuredangle(u_\star, x_2^\dagger) - \sin \measuredangle(x_2^\dagger, x_3^\dagger),$$

for small enough $\eta$ and $\lambda$ we have that Proposition 1 holds, $\zeta > 0$, and it is straightforward to check that $\zeta/\|x_2^\dagger\| < z_2$.

The main idea here is that the properties of the dataset, in particular the inequality in Proposition 1 (c), allow us to express $u_\star$ as the sum of two neurons, where the first is close to $u_\star$ and the second is crafted using a shortening by projection whose validity relies on the ReLU non-linearity. Specifically, we define:

$$u_1 := u_\star - \zeta\,\overline{x}_2^\dagger \qquad\qquad u_2 := \zeta(\overline{x}_2^\dagger - \overline{x}_3^\dagger \overline{x}_3^{\dagger\top} \overline{x}_2^\dagger).$$

Next we define $\theta = (W, a) \in \mathbb{R}^{m\times(d+1)}$ by expressing $u_1$ and $u_2$ using balanced inner and output layers:

$$w_i := \frac{1}{\sqrt{\|u_i\|}} u_i \text{ and } a_i := \sqrt{\|u_i\|} \text{ for both } i \in \{1, 2\}$$
$$w_i := 0 \text{ and } a_i := 0 \text{ for all } i > 2.$$

Recalling that $x_k^{\dagger\top} x_{k'}^\dagger = \mathbb{1}(k = k')$ for all $k, k' \in [d]$ by the definition of $(x_k^\dagger)_{k\in[d]}$, and that $x_3^{\dagger\top} x_2^\dagger > 0$ for small enough $\eta$, we have that, on all training points, the network $\theta$ agrees with every network in $\Theta_{u_\star}$:

$$f_\theta(x_2) = \sigma(u_1^\top x_2) + \sigma(u_2^\top x_2) = (z_2 - \zeta/\|x_2^\dagger\|) + \zeta/\|x_2^\dagger\| = z_2$$
$$f_\theta(x_3) = \sigma(u_1^\top x_3) + \sigma(u_2^\top x_3) = z_3 + \sigma(-(\zeta/\|x_3^\dagger\|)\,\overline{x}_3^{\dagger\top} \overline{x}_2^\dagger) = z_3$$
$$f_\theta(x_k) = u_\star^\top x_k = z_k \quad \text{for all } k \notin \{2, 3\}.$$

Now we verify that the network $\theta$ has smaller norm:

$$\|\theta\|^2 = \textstyle\sum_{i\in\{1,2\}} (a_i^2 + \|w_i\|^2)$$
$$= 2(\|u_1\| + \|u_2\|)$$
$$= 2\left(\sqrt{\|u_\star\|^2 + \zeta^2 - 2\zeta\cos\measuredangle(u_\star, x_2^\dagger)} + \zeta\sin\measuredangle(x_2^\dagger, x_3^\dagger)\right)$$
$$\leq 2\left(1/2 + \|u_\star\|^2/2 + \zeta\big(\zeta/2 - \cos\measuredangle(u_\star, x_2^\dagger) + \sin\measuredangle(x_2^\dagger, x_3^\dagger)\big)\right)$$
$$= 1 + \|u_\star\|^2 - \zeta^2$$

$$\leq 2\|u_\star\| - \zeta^2/2,$$

where the last inequality holds for small enough $\lambda$.

Hence

$$
\begin{aligned}
L_\lambda(\theta) &= \frac{1}{d} \sum_{k=1}^{d} (f_\theta(x_k) - y_k)^2 + \lambda \|\theta\|^2 \\
&\leq \|u_\star - v_\star\|_H^2 + 2\lambda\|u_\star\| - \lambda\zeta^2/2 \\
&< L_\lambda^\star.
\end{aligned}
$$
□

Now we turn to considering a subgradient flow as in Equation (3), beginning with a proposition about the random initialization according to Assumption 2 (a). To state it, we define notations for the sets of indices of training points that are on the boundary or strictly inside of the active half-space of a ReLU neuron $w \in \mathbb{R}^d$:

$$K_0(w) \coloneqq \{k \in [d] \mid w^\top x_k = 0\} \qquad\qquad K_+(w) \coloneqq \{k \in [d] \mid w^\top x_k > 0\};$$

define notations $\mathsf{s}_i$ for the signs of the output weights (which we shall shortly show stay unchanged throughout the training), and $I_+$ for the set of indices of neurons that have positive output weight and are initially active on at least one training point:

$$\mathsf{s}_i \coloneqq \operatorname{sign}(a_i(0)) \qquad\qquad I_+ \coloneqq \{i \in [m] \mid \mathsf{s}_i = 1 \text{ and } K_+(w_i(0)) \neq \emptyset\};$$

and define the vector that will be the focus of the early alignment:

$$\gamma \coloneqq \frac{2}{d} \sum_{k \in [d]} y_k\, x_k.$$

The proposition lower bounds the probability that the initialization has the following regularity properties: the set $I_+$ just defined is non-empty, no neuron is exactly orthogonal to some training point, and no neuron with negative output weight has exactly the same direction as the vector $\gamma$.

**Proposition 3.** *With probability at least* $1 - (\frac{3}{4})^m$, *we have:* $I_+ \neq \emptyset$, $K_0(w_i(0)) = \emptyset$ *for all* $i \in [m]$, *and* $\measuredangle(w_i(0), \gamma) > 0$ *for all* $i \in [m]$ *with* $\mathsf{s}_i = -1$.

*Proof.* The $2m$ events $\mathsf{s}_i = 1$ and $K_+(w_i(0)) \neq \emptyset$ are independent and have probability at least $\frac{1}{2}$, so the probability that $I_+ = \emptyset$ is at most $(\frac{3}{4})^m$. The events $K_0(w_i(0)) \neq \emptyset$ and $\measuredangle(w_i(0), \gamma) = 0$ all have probability 0. □

In the next proposition, part (a) states that the balancedness between the inner and output layers at initialization is preserved throughout the training, and that the output layer signs do not change; and part (b) spells out the time derivatives of the hidden neurons and their normalized versions, as well as of the output weights and the logarithms of their absolute values.

**Proposition 4.**     *(a) For all* $i \in [m]$ *and all* $t \in \mathbb{R}_+$ *we have* $a_i(t) = \mathsf{s}_i\|w_i(t)\|$.

   *(b) For all* $i \in [m]$ *and almost all* $t \in \mathbb{R}_+$ *we have:*

$$\dot{w}_i(t) = a_i(t)\, g_i(t) - 2\lambda w_i(t) \qquad\qquad \dot{\overline{w}}_i(t) = \mathsf{s}_i\big(g_i(t) - \overline{w}_i(t)\, \overline{w}_i(t)^\top g_i(t)\big)$$

$$\dot{a}_i(t) = w_i(t)^\top g_i(t) - 2\lambda a_i(t) \qquad\qquad \frac{\mathrm{d}(\ln\|w_i(t)\|)}{\mathrm{d}t} = \mathsf{s}_i\, \overline{w}_i(t)^\top g_i(t) - 2\lambda,$$

   *where*

$$g_i(t) \in \frac{2}{d} \sum_{k \in [d]} (y_k - f_{\theta(t)}(x_k))\partial\sigma(w_i(t)^\top x_k)x_k.$$

*Proof.* Since, in our setting of two-layer ReLU networks and regularized square loss, the chain rule applies (see, e.g., Davis et al., 2020, Theorem 5.8), the equations for $\dot{w}_i(t)$ and $\dot{a}_i(t)$ in part (b) follow by straightforward expansions.

Now, for all $i \in [m]$ and almost all $t \in \mathbb{R}_+$ we have

$$\mathrm{d}(a_i^2(t) - \|w_i(t)\|^2)/\mathrm{d}t = -4\lambda(a_i^2(t) - \|w_i(t)\|^2),$$

and recall that we have balancedness at the initialization, i.e., $a_i(0) = \mathsf{s}_i\|w_i(0)\|$. Hence, for part (a), it remains to show that each $a_i(t)$ maintains its initial sign, i.e., remains non-zero. That follows by the next lower bound on the derivative of $\ln|a_i(t)|$, where the last inequality is a consequence of the loss being non-increasing (see, e.g., Davis et al., 2020, Lemma 5.2):

$$\frac{\mathrm{d}|\ln a_i(t)|}{\mathrm{d}t} \geq -\|g_i(t)\| - 2\lambda \geq -\frac{2}{d}\sum_{k \in [d]}|y_k - f_{\theta(t)}(x_k)| - 2\lambda \geq -2\sqrt{L_\lambda(\theta(t))} - 2\lambda \geq -2\sqrt{L_\lambda(\theta(0))} - 2\lambda.$$

The remainder of part (b), namely the equations for $\dot{\overline{w}}_i(t)$ and $\mathrm{d}(\ln\|w_i(t)\|)/\mathrm{d}t$, now follow by straightforward calculations. $\qquad\square$

At this moment we are equipped for the first of three lemmas, which will contain our analysis of the training split into three phases. It describes the state of the network at time

$$T_1 := \epsilon \ln(1/\alpha)/\|\gamma\|,$$

which we regard as the end of the early alignment phase. Namely, all neurons with positive output weight and that were initially active on at least one training point are still small, active on all training points, and closely aligned to the vector $\gamma$; and all other neurons are even smaller, and for the remainder of the training remain deactivated from all training points and shrink by the weight decay.

**Lemma 3.** *Provided the initialization has the properties in Proposition 3, and provided $\eta$ and $\alpha$ are sufficiently small, the following hold.*

*(a) For all $i \in I_+$ we have:*

$$\|w_i(T_1)\| < 2\alpha^{1-\epsilon} \qquad\qquad K_+(w_i(T_1)) = [d] \qquad\qquad \overline{w}_i(T_1)^\top \overline{\gamma} \geq 1 - \alpha^\epsilon.$$

*(b) For all $i \in [m] \setminus I_+$ and all $t \geq T_1$ we have:*

$$\|w_i(T_1)\| \leq \alpha^{1+2\lambda\epsilon/\|\gamma\|} \qquad\qquad K_+(w_i(t)) = \emptyset \qquad\qquad \|w_i(t)\| = e^{-2\lambda(t-T_1)}\|w_i(T_1)\|.$$

*Proof.* First we show the next claim, which has as a corollary the assertion $\|w_i(T_1)\| < 2\alpha^{1-\epsilon}$ in part (a).

**Claim 1.** *Provided $\alpha$ is sufficiently small, for all $i \in [m]$ and all $t \in [0, T_1]$ we have $\|w_i(t)\| < 2\alpha^{1-\epsilon t/T_1}$.*

*Proof of Claim 1.* For a contradiction, suppose

$$t = \inf\{t \in [0, T_1] \mid \|w_i(t)\| \geq 2\alpha^{1-\epsilon t/T_1} \text{ for some } i \in [m]\}.$$

Since each $w_i(t)$ is continuous, there exists $i \in [m]$ such that $\|w_i(t)\| = 2\alpha^{1-\epsilon t/T_1}$, and for all $j \in [m]$ and all $\tau \in [0, t]$ we have $\|w_j(\tau)\| \leq 2\alpha^{1-\epsilon\tau/T_1}$. We then observe that

$$\|w_i(t)\| \leq \alpha \exp(t \max_{\tau \in [0,t]} \|g_i(\tau)\|) \qquad\qquad \text{by Proposition 4 (b), Grönwall's inequality}$$

$$\leq \alpha \exp\left(t\big(\|\gamma\| + 2 \max_{\substack{k \in [d] \\ \tau \in [0,t]}} |f_{\theta(\tau)}(x_k)|\big)\right) \qquad \text{by Proposition 4 (b) and } \|x_k\| = 1$$

$$\leq \alpha \exp\!\Big(t\big(\|\gamma\| + 2m \max_{\substack{j\in[m]\\ \tau\in[0,t]}} \|w_j(\tau)\|^2\big)\Big) \qquad \text{by Equation (1) and Proposition 4 (a)}$$

$$\leq \alpha \exp\!\Big(t\big(\|\gamma\| + 8m \max_{\tau\in[0,t]} \alpha^{2-2\epsilon\tau/T_1}\big)\Big) \qquad \text{since } \|w_j(\tau)\| \leq 2\alpha^{1-\epsilon\tau/T_1}$$

$$\leq \alpha \exp\!\big(t(\|\gamma\| + 8m\,\alpha^{2-2\epsilon})\big) \qquad \text{since } \alpha \leq 1 \text{ and } \tau \leq T_1$$

$$\leq \alpha \exp(t\|\gamma\| + 8mT_1\alpha^{2-2\epsilon}) \qquad \text{since } t \leq T_1$$

$$= \alpha^{1-\epsilon t/T_1 - 8m\epsilon\,\alpha^{2-2\epsilon}/\|\gamma\|} \qquad \text{since } \exp(T_1\|\gamma\|) = \alpha^{-\epsilon}$$

$$< \alpha^{1-\epsilon t/T_1 - (\ln 2)/\ln(1/\alpha)} \qquad \text{for small enough } \alpha$$

$$= 2\alpha^{1-\epsilon t/T_1},$$

which contradicts $\|w_i(t)\| = 2\alpha^{1-\epsilon t/T_1}$. $\qquad\qquad\square$

Second we show a claim from which $\|w_i(T_1)\| \leq \alpha^{1+2\lambda\epsilon/\|\gamma\|}$ follows by recalling that $e^{T_1\|\gamma\|} = \alpha^{-\epsilon}$.

**Claim 2.** *Provided $\eta$ and $\alpha$ are sufficiently small, with probability $1$, for all $i \in [m] \setminus I_+$ and almost all $t \in [0, T_1]$ we have $\mathrm{d}(\ln\|w_i(t)\|)/\mathrm{d}t \leq -2\lambda$.*

*Proof of Claim 2.* Consider $i \in [m] \setminus I_+$.

If $K_+(w_i(0)) = \emptyset$, we can assume that also $K_0(w_i(0)) = \emptyset$, which occurs with probability $1$. Then from Proposition 4 (b) for all $t \in \mathbb{R}_+$ we have $\dot{\overline{w}}_i(t) = 0$ and $\mathrm{d}(\ln\|w_i(t)\|)/\mathrm{d}t = -2\lambda$, i.e., hidden neuron $w_i(t)$ has constant direction and its length decreases exponentially at constant rate $2\lambda$.

Otherwise, we have $\mathsf{s}_i = -1$. Consider $t \in [0, T_1]$ for which Proposition 4 (b) applies. By the equation for $\mathrm{d}(\ln\|w_i(t)\|)/\mathrm{d}t$, it suffices to show that $w_i(t)^\top g_i(t) \geq 0$. But since

$$w_i(t)^\top g_i(t) = \frac{2}{d} \sum_{k\in[d]} (y_k - f_{\theta(t)}(x_k))\sigma(w_i(t)^\top x_k) = \frac{2}{d} \sum_{k\in K_+(w_i(t))} (y_k - f_{\theta(t)}(x_k))w_i(t)^\top x_k,$$

it suffices to verify that $f_{\theta(t)}(x_k) \leq y_k$ for all $k \in [d]$, which follows Proposition 1 (b) and Claim 1 for small enough $\alpha$. $\qquad\square$

Now observe that the second assertion $K_+(w_i(T_1)) = [d]$ in part (a) follows from the third one $\overline{w}_i(T_1)^\top \overline{\gamma} \geq 1 - \alpha^\epsilon$ and Proposition 1 (b) for small enough $\alpha$.

Next observe that the remainder of part (b) (i.e., that we have $K_+(w_i(t)) = \emptyset$ and $\|w_i(t)\| = e^{-2\lambda(t-T_1)}\|w_i(T_1)\|$ for all $t \geq T_1$) follows from: $K_+(w_i(T_1)) = \emptyset$, Assumption 2 (b), Proposition 4 (a), and the equation for $\dot{a}_i(t)$ in Proposition 4 (b).

Therefore it remains to establish that $\overline{w}_i(T_1)^\top \overline{\gamma} \geq 1 - \alpha^\epsilon$ for all $i \in I_+$, and that $K_+(w_i(T_1)) = \emptyset$ for all $i \in [m] \setminus I_+$. These follow by the proofs of Chistikov et al. (2023, Lemmas 3 and 5, and Proposition 18), which carry over to our setting without significant modifications once Chistikov et al. (2023, Lemma 19) is replaced by Claims 1 and 2 above. The latter is the only part affected by the regularization, i.e., by the presence of the $-2\lambda$ term in the equation for $\mathrm{d}(\ln\|w_i(t)\|)/\mathrm{d}t$ in Proposition 4 (b); the remainder of the reasoning is based on the equation for $\dot{\overline{w}}_i(t)$, which is the same with and without the regularization. Note also that Chistikov et al. (2023, Assumption 1) is satisfied due to Assumptions 1 and 2, Proposition 1 (a), and Proposition 3 above. $\qquad\square$

Now we come to the second lemma, which handles the intermediate phase of the training that begins when the early alignment ends at time $T_1$. At times $t \geq T_1$, a key role will be played by the vector

$$v(t) := \sum_{i\in I_+} a_i(t)\, w_i(t)$$

obtained by composing the aligned neurons. Again, the lemma describes the state of the network at the end of the phase. Namely, at some time $T_2$, the composite vector $v(T_2)$ will be near to the teacher vector $v_\star$, and all the constituent neurons will still be closely aligned (no longer with the vector $\gamma$ but with each other).

**Lemma 4.** *Provided the initialization has the properties in Proposition 3, and provided $\eta$ and $\alpha$ are sufficiently small and $\lambda \leq \mu_{\min}\alpha^{2\epsilon}$, there exists $T_2 > T_1$ such that:*

(a) $\|v_\star - v(T_2)\| \leq \alpha^\epsilon$.

(b) $\overline{w}_i(T_2)^\top \overline{w}_{i'}(T_2) \geq 1 - 4\alpha^\epsilon$ *for all* $i, i' \in I_+$.

*Proof.* Let:

$$g(t) := 2H(v_\star - v(t)) \qquad\qquad \delta(t) := \|v(t)\|\big(g(t) + \overline{v}(t)\overline{v}(t)^\top g(t)\big).$$

By Lemma 3 and the proof of Chistikov et al. (2023, Theorem 6), the time

$$T_2 := \inf\{t \geq T_1 \mid \|\dot{v}(t) - \delta(t)\| \geq 3\alpha^{\epsilon/2}\|\delta(t)\| \text{ or } \|v_\star - v(t)\| \leq \alpha^\epsilon\}$$

is finite, and moreover for all $t \in [T_1, T_2]$, all $i, i' \in I_+$, and all $k \in [d]$ we have:

$$\overline{v}(t)^\top x_k > \sqrt{8}\alpha^{\epsilon/2} \qquad\qquad \overline{w}_i(t)^\top \overline{w}_{i'}(t) > 1 - 4\alpha^\epsilon \qquad\qquad \overline{v}(t)^\top \overline{g}(t) > \alpha^{\epsilon/3}.$$

Consequently, since $\sin\xi \leq \sqrt{2(1 - \cos\xi)}$, for all $t \in [T_1, T_2]$, all $i \in I_+$, and all $k \in [d]$, we also have:

$$\overline{w}_i(t)^\top x_k > 0 \qquad f_{\theta(t)}(x_k) = v(t)^\top x_k \qquad g_i(t) = g(t) \qquad \overline{w}_i(t)^\top \overline{g}(t) > \alpha^{\epsilon/3}/2$$

and $v(t)$ is differentiable.

We already have part (b). To establish part (a), it suffices to show that $\|\dot{v}(T_2) - \delta(T_2)\| < 3\alpha^{\epsilon/2}\|\delta(T_2)\|$. We reason as follows, where the argument $T_2$ of all vectors is omitted for readability:

$$\|\dot{v} - \delta\| = \left\|\sum_{i \in I_+} \|w_i\|^2\big(g + \overline{w}_i\overline{w}_i^\top g - 4\lambda\overline{w}_i\big) - \delta\right\| \qquad \text{by Proposition 4 and } g_i = g$$

$$= \left\|\sum_{i \in I_+} \|w_i\|^2\big(g + \overline{w}_i\overline{w}_i^\top g - 4\lambda\overline{w}_i\big)\right.$$

$$\left. - \sum_{i \in I_+} \|w_i\|^2\big((\overline{v}^\top\overline{w}_i)g + \overline{v}\,\overline{w}_i^\top g\big)\right\| \qquad \text{since } v = \sum_{i \in I_+} \overline{v}\,\overline{v}^\top a_i\, w_i$$

$$= \left\|\sum_{i \in I_+} \|w_i\|^2\big((1 - \overline{v}^\top\overline{w}_i)g - 4\lambda\overline{w}_i\big)\right.$$

$$\left. + \sum_{i \in I_+} \|w_i\|^2(\overline{w}_i - \overline{v})\overline{w}_i^\top g\right\| \qquad \text{by rearranging}$$

$$\leq \sum_{i \in I_+} \|w_i\|^2\big(|1 - \overline{v}^\top\overline{w}_i|\,\|g\| + 4\lambda\big) \qquad \text{by the triangle inequality}$$

$$+ \sum_{i \in I_+} \|w_i\|^2\|\overline{w}_i - \overline{v}\|\,\overline{w}_i^\top g \qquad \text{and } \overline{w}_i^\top g \geq 0$$

$$\leq (4\alpha^\epsilon\|g\| + 4\lambda)\sum_{i \in I_+} \|w_i\|^2 \qquad \text{since } \overline{v}^\top\overline{w}_i \geq 1 - 4\alpha^\epsilon$$

$$+ \sqrt{8}\alpha^{\epsilon/2}\sum_{i \in I_+} \|w_i\|^2\,\overline{w}_i^\top g \qquad \text{and } \sin\xi \leq \sqrt{2(1 - \cos\xi)}$$

$$= (4\alpha^\epsilon\|g\| + 4\lambda)\sum_{i \in I_+} \|w_i\|^2 + \sqrt{8}\alpha^{\epsilon/2}\,v^\top g \qquad \text{since } v = \sum_{i \in I_+} \|w_i\|^2\,\overline{w}_i$$

$$\leq \frac{4\alpha^\epsilon \|g\| + 4\lambda}{1 - 4\alpha^\epsilon} \|v\| + \sqrt{8}\alpha^{\epsilon/2} \, v^\top g \qquad\qquad \text{since } \overline{v}^\top \overline{w}_i \geq 1 - 4\alpha^\epsilon$$

$$\leq \frac{4\alpha^\epsilon (\|g\| + \mu_{\min}\alpha^\epsilon)}{1 - 4\alpha^\epsilon} \|v\| + \sqrt{8}\alpha^{\epsilon/2} \, v^\top g \qquad\qquad \text{since } \lambda \leq \mu_{\min}\alpha^{2\epsilon}$$

$$\leq \frac{6\alpha^\epsilon}{1 - 4\alpha^\epsilon} \|v\|\|g\| + \sqrt{8}\alpha^{\epsilon/2} \, v^\top g \qquad\qquad \text{since } \|g\| \geq 2\mu_{\min}\|v_\star - v\|$$

$$\leq \left( \frac{6\alpha^\epsilon}{1 - 4\alpha^\epsilon} + \sqrt{8}\alpha^{\epsilon/2} \right)\|\delta\| \qquad\qquad \text{since } \angle(g, \overline{v}\,\overline{v}^\top g) \leq \pi/2$$

$$< 3\alpha^{\epsilon/2}\|\delta\| \qquad\qquad \text{for small enough } \alpha$$

and since $\|v\| > 0$ by Proposition 4 (a). $\qquad\qquad\qquad\qquad\qquad\qquad\qquad\qquad\qquad\qquad$ □

Finally, our third lemma gives an account of the late convergence phase that begins when the intermediate phase ends at time $T_2$. It establishes that the subgradient flow converges to a network in $\Theta_{u_\star}$, i.e., a rank-1 minimizer of the regularized loss $L_\lambda$.

**Lemma 5.** *Provided the initialization has the properties in Proposition 3, and provided $\eta$ and $\alpha$ are sufficiently small and $\lambda \leq \mu_{\min}\alpha^{2\epsilon}$, we have that $\lim_{t\to\infty} \theta(t) \in \Theta_{u_\star}$.*

*Proof.* Let

$$T_3 := \inf\{t \geq T_2 \mid \overline{w}_i(t)^\top \overline{w}_{i'}(t) \leq 1 - \alpha^{\epsilon/2} \text{ for some } i, i' \in I_+\}.$$

Thus, recalling Lemma 4 (a) by which the composite vector $v(t)$ at time $t = T_2$ is within $\alpha^\epsilon$ of the teacher vector $v_\star$, and also Lemma 3 (b) by which the neurons $w_i(t)$ with $i \notin I_+$ at all times $t \geq T_1$ do not contribute to the network outputs $f_{\theta(t)}(x_k)$ for any $k \in [d]$: for small enough $\alpha$, all $t \in [T_2, T_3)$, all $i, i' \in I_+$, and all $k \in [d]$ we have $\overline{w}_i(t)^\top x_k > 0$ and $f_{\theta(t)}(x_k) = v(t)^\top x_k$, hence

$$L_\lambda(\theta(t)) = \|v_\star - v(t)\|_H^2 + 2\lambda \sum_{i \in [m]} \|w_i(t)\|^2, \tag{8}$$

so if $i \in I_+$ then

$$-\frac{1}{2}\nabla_{w_i(t)} L_\lambda = (\nabla_{w_i(t)} v(t))^\top H(v_\star - v(t)) - \lambda w_i(t)$$

$$= a_i(t) H(v_\star - v(t)) - \lambda w_i(t)$$

$$= a_i(t) H(v_\star - u_\star) + a_i(t) H(u_\star - v(t)) - \lambda w_i(t)$$

$$= a_i(t)\big(H(u_\star - v(t)) + \lambda(\overline{u}_\star - \overline{w}_i(t))\big)$$

$$-\frac{1}{2}\nabla_{a_i(t)} L_\lambda = (\nabla_{a_i(t)} v(t))^\top H(v_\star - v(t)) - \lambda a_i(t)$$

$$= w_i(t)^\top H(v_\star - v(t)) - \lambda a_i(t)$$

$$= w_i(t)^\top H(v_\star - u_\star) + w_i(t)^\top H(u_\star - v(t)) - \lambda a_i(t)$$

$$= w_i(t)^\top \big(H(u_\star - v(t)) + \lambda(\overline{u}_\star - \overline{w}_i(t))\big),$$

and if $i \in [m] \setminus I_+$ then

$$-\frac{1}{2}\nabla_{w_i(t)} L_\lambda = -\lambda w_i(t) \qquad\qquad\qquad -\frac{1}{2}\nabla_{a_i(t)} L_\lambda = -\lambda a_i(t),$$

therefore putting the two cases together we have

$$-\frac{1}{2}\nabla_{w_i(t)} L_\lambda = a_i(t) h_i(t) \qquad\qquad\qquad -\frac{1}{2}\nabla_{a_i(t)} L_\lambda = w_i(t)^\top h_i(t) \tag{9}$$

where

$$h_i(t) := \begin{cases} H(u_\star - v(t)) + \lambda(\overline{u}_\star - \overline{w}_i(t)) & \text{if } i \in I_+, \\ -\lambda \overline{w}_i(t) & \text{if } i \in [m] \setminus I_+. \end{cases} \tag{10}$$

Consider $t \in [T_2, T_3)$, and let $\mu_{\max}$ denote the largest eigenvalue of $H$.

For small enough $\alpha$, by Lemma 4 we have

$$L_\lambda(\theta(T_2)) \leq \mu_{\max}\alpha^{2\epsilon} + 2\mu_{\min}\alpha^{2\epsilon}\frac{1+\alpha^\epsilon}{1-4\alpha^\epsilon} \leq (\mu_{\max} + 3\mu_{\min})\alpha^{2\epsilon}, \tag{11}$$

so by the loss being non-increasing (see, e.g., Davis et al., 2020, Lemma 5.2) and Equation (8) we have

$$\|v_\star - v(t)\| \leq \sqrt{\frac{L_\lambda(\theta(t))}{\mu_{\min}}} \leq \sqrt{\frac{L_\lambda(\theta(T_2))}{\mu_{\min}}} \leq \sqrt{\left(\frac{\mu_{\max}}{\mu_{\min}} + 3\right)\alpha^{2\epsilon}} < \alpha^{\epsilon/2}. \tag{12}$$

Thus, at each time $t$ in this phase, we not only have that the $I_+$ neurons are closely aligned (by the definition of time $T_3$ at the start of the current proof) but also that their composite vector $v(t)$ is inside a small ball around the teacher vector $v_\star$ (which also contains the claimed limit $u_\star$).

Letting $Q_i(t) := \|w_i(t)\|^2(I_d + \overline{w}_i(t)\overline{w}_i(t)^\top)$ for all $i \in [m]$, we have:

$$\left\|\frac{1}{2}\nabla L_\lambda(\theta(t))\right\|^2 = \sum_{i \in I_+} \|H(u_\star - v(t)) + \lambda(\overline{u}_\star - \overline{w}_i(t))\|^2_{Q_i(t)} \qquad \text{by Equations (9) and (10)}$$

$$+ \sum_{i \in [m]\setminus I_+} \|\lambda\overline{w}_i(t)\|^2_{Q_i(t)} \qquad \text{and Equation (10)}$$

$$\geq \sum_{i \in I_+} \|w_i(t)\|^2\|H(u_\star - v(t)) + \lambda(\overline{u}_\star - \overline{w}_i(t))\|^2 \qquad \text{by omitting}$$

$$+ \sum_{i \in [m]\setminus I_+} \|w_i(t)\|^2\|\lambda\overline{w}_i(t)\|^2 \qquad \left\|\frac{1}{2}\nabla_{a_i(t)}L_\lambda\right\|^2 \text{ terms}$$

$$= \sum_{i \in I_+} \|w_i(t)\|^2\|H^{\frac{1}{2}}(u_\star - v(t)) + \lambda H^{-\frac{1}{2}}(\overline{u}_\star - \overline{w}_i(t))\|^2_H \qquad \text{since } H \text{ is pos. def.}$$

$$+ \lambda^2 \sum_{i \in [m]\setminus I_+} \|w_i(t)\|^2 \qquad \text{and by simplifying}$$

$$\geq \mu_{\min} \sum_{i \in I_+} \|w_i(t)\|^2\|H^{\frac{1}{2}}(u_\star - v(t)) + \lambda H^{-\frac{1}{2}}(\overline{u}_\star - \overline{w}_i(t))\|^2$$

$$+ \lambda^2 \sum_{i \in [m]\setminus I_+} \|w_i(t)\|^2 \qquad \text{since } \|\cdot\|^2_H \geq \mu_{\min}\|\cdot\|^2$$

$$= \mu_{\min}\left(\sum_{i \in I_+} \|w_i(t)\|^2\right)\|u_\star - v(t)\|^2_H \qquad \text{expanding the square}$$

$$+ 2\mu_{\min}\lambda(u_\star - v(t))^\top\left(\sum_{i \in I_+} \|w_i(t)\|^2\overline{u}_\star - v(t)\right) \qquad \text{and recalling that,}$$

$$+ \mu_{\min}\lambda^2 \sum_{i \in I_+} \|w_i(t)\|^2\|\overline{u}_\star - \overline{w}_i(t)\|^2_{H^{-1}} \qquad \text{by Proposition 4 (a),}$$

$$+ \lambda^2 \sum_{i \in [m]\setminus I_+} \|w_i(t)\|^2 \qquad v(t) = \sum_{i \in I_+} \|w_i(t)\|^2\overline{w}_i(t)$$

$$\geq \mu_{\min}\left(\sum_{i \in I_+} \|w_i(t)\|^2\right)\|u_\star - v(t)\|^2_H \qquad \text{by decomposing } v(t)$$

$$+ 2\mu_{\min}\lambda(u_\star - \overline{u}_\star\overline{u}_\star^\top v(t))^\top\left(\sum_{i \in I_+} \|w_i(t)\|^2\overline{u}_\star - \overline{u}_\star\overline{u}_\star^\top v(t)\right) \qquad \text{as } v(t) - \overline{u}_\star\overline{u}_\star^\top v(t)$$

$$+ \frac{\mu_{\min}}{\mu_{\max}}\lambda^2 \sum_{i \in I_+} \|w_i(t)\|^2\|\overline{u}_\star - \overline{w}_i(t)\|^2 \qquad \text{plus } \overline{u}_\star\overline{u}_\star^\top v(t),$$

$$+ \lambda^2 \sum_{i \in [m] \backslash I_+} \|w_i(t)\|^2 \qquad \text{and } \| \cdot \|_{H^{-1}}^2 \geq \mu_{\max}^{-1} \| \cdot \|^2$$

$$= \mu_{\min} \left( \sum_{i \in I_+} \|w_i(t)\|^2 \right) \|u_\star - v(t)\|_H^2 \qquad \text{since } \overline{u}_\star^\top \overline{u}_\star = 1,$$

$$+ 2\mu_{\min}\lambda(\|u_\star\| - \overline{u}_\star^\top v(t))\big( \sum_{i \in I_+} \|w_i(t)\|^2 - \overline{u}_\star^\top v(t) \big) \qquad \|\overline{u}_\star - \overline{w}_i(t)\|^2$$

$$+ 2\frac{\mu_{\min}}{\mu_{\max}}\lambda^2 \big( \sum_{i \in I_+} \|w_i(t)\|^2 - \overline{u}_\star^\top v(t) \big) \qquad = 2(1 - \overline{u}_\star^\top \overline{w}_i(t)), \text{ and}$$

$$+ \lambda^2 \sum_{i \in [m] \backslash I_+} \|w_i(t)\|^2 \qquad v(t) = \sum_{i \in I_+} \|w_i(t)\|^2 \, \overline{w}_i(t).$$

To justify in greater detail the last inequality, writing $v_\|(t) := \overline{u}_\star \overline{u}_\star^\top v(t)$ and $v_\perp(t) := v(t) - v_\|(t)$, observe that

$$(u_\star - v(t))^\top \big( \sum_{i \in I_+} \|w_i(t)\|^2 \, \overline{u}_\star - v(t) \big) = \big( (u_\star - v_\|(t)) - v_\perp(t) \big)^\top \Big( \big( \sum_{i \in I_+} \|w_i(t)\|^2 \, \overline{u}_\star - v_\|(t) \big) - v_\perp(t) \Big)$$

$$= (u_\star - v_\|(t))^\top \big( \sum_{i \in I_+} \|w_i(t)\|^2 \, \overline{u}_\star - v_\|(t) \big) + \|v_\perp(t)\|^2$$

$$\geq (u_\star - v_\|(t))^\top \big( \sum_{i \in I_+} \|w_i(t)\|^2 \, \overline{u}_\star - v_\|(t) \big).$$

Now, if $\|u_\star\| - \overline{u}_\star^\top v(t) \geq -\frac{\lambda}{2\mu_{\max}}$ then

$$2\mu_{\min}\lambda(\|u_\star\| - \overline{u}_\star^\top v(t))\big( \sum_{i \in I_+} \|w_i(t)\|^2 - \overline{u}_\star^\top v(t) \big) \geq -\frac{\mu_{\min}}{\mu_{\max}}\lambda^2 \big( \sum_{i \in I_+} \|w_i(t)\|^2 - \overline{u}_\star^\top v(t) \big),$$

and if $\|u_\star\| - \overline{u}_\star^\top v(t) < -\frac{\lambda}{2\mu_{\max}}$ then

$$2\mu_{\min}\lambda(\|u_\star\| - \overline{u}_\star^\top v(t))\big( \sum_{i \in I_+} \|w_i(t)\|^2 - \overline{u}_\star^\top v(t) \big)$$

$$\geq -4\mu_{\min}\mu_{\max}(\|u_\star\| - \overline{u}_\star^\top v(t))^2 \big( \sum_{i \in I_+} \|w_i(t)\|^2 - \overline{u}_\star^\top v(t) \big)$$

$$\geq -4\mu_{\min}\mu_{\max}\|u_\star - v(t))\|^2 \big( \sum_{i \in I_+} \|w_i(t)\|^2 - \overline{u}_\star^\top v(t) \big)$$

$$\geq -4\mu_{\max}\|u_\star - v(t))\|_H^2 \big( \sum_{i \in I_+} \|w_i(t)\|^2 - \overline{u}_\star^\top v(t) \big)$$

$$\geq -\frac{1}{2}\mu_{\min}\big( \sum_{i \in I_+} \|w_i(t)\|^2 \big) \|u_\star - v(t)\|_H^2$$

since $1 - \frac{\overline{u}_\star^\top v(t)}{\sum_{i \in I_+} \|w_i(t)\|^2} \leq \frac{\mu_{\min}}{8\mu_{\max}}$ for small enough $\alpha$ by Equation (12).

Therefore, in either case, for small enough $\alpha$ we have

$$\left\| \frac{1}{2}\nabla L_\lambda(\theta(t)) \right\|^2 \geq \frac{1}{2}\mu_{\min}\big( \sum_{i \in I_+} \|w_i(t)\|^2 \big) \|u_\star - v(t)\|_H^2$$

$$+ \frac{\mu_{\min}}{\mu_{\max}}\lambda^2 \big( \sum_{i \in I_+} \|w_i(t)\|^2 - \overline{u}_\star^\top v(t) \big)$$

$$+ \lambda^2 \sum_{i \in [m] \setminus I_+} \|w_i(t)\|^2$$

$$\geq \frac{1}{4} \mu_{\min} \|u_\star - v(t)\|_H^2$$

$$+ \frac{\mu_{\min}}{\mu_{\max}} \lambda^2 \Big( \sum_{i \in [m]} \|w_i(t)\|^2 - \overline{u}_\star^\top v(t) \Big). \tag{13}$$

Using the lower bound on the square of the gradient in Equation (13), which we obtained by distinguishing the two cases depending on whether the projection of $v(t)$ onto the direction of $u_\star$ significantly exceeds the length of $u_\star$, we are now able to show a local Polyak-Łojasiewicz inequality. However, we first need to adjust the regularized loss by subtracting from it the value of each network in the set $\Theta_{u_\star}$ (see Proposition 2 (a)).

Letting

$$\widehat{L}_\lambda(\theta) := L_\lambda(\theta) - L_\lambda^\star,$$

observe that:

$$
\begin{aligned}
\widehat{L}_\lambda(\theta(t)) &= \|v_\star - v(t)\|_H^2 - \|v_\star - u_\star\|_H^2 \\
&\quad + 2\lambda \Big( \sum_{i \in [m]} \|w_i(t)\|^2 - \|u_\star\| \Big) &&\text{by Equation (8)} \\
&= \|v_\star - v(t)\|_H^2 - \|v_\star - u_\star\|_H^2 \\
&\quad + 2 \Big( \sum_{i \in [m]} \|w_i(t)\|^2 - \|u_\star\| \Big) \|H(v_\star - u_\star)\| &&\text{by Equation (7)} \\
&= \|u_\star - v(t)\|_H^2 &&\text{by expanding} \\
&\quad + 2(u_\star - v(t))^\top H(v_\star - u_\star) &&\|v_\star - v(t)\|_H^2 = \\
&\quad + 2 \Big( \sum_{i \in [m]} \|w_i(t)\|^2 - \|u_\star\| \Big) \|H(v_\star - u_\star)\| &&\|(u_\star - v(t)) + (v_\star - u_\star)\|_H^2 \\
&= \|u_\star - v(t)\|_H^2 &&\text{by Equation (7)} \\
&\quad + 2\lambda \Big( \sum_{i \in [m]} \|w_i(t)\|^2 - \overline{u}_\star^\top v(t) \Big) &&\text{and simplifying} \tag{14} \\
&\leq \frac{1}{\kappa} \|\nabla \widehat{L}_\lambda(\theta(t))\|^2 &&\text{by Equation (13)}
\end{aligned}
$$

where $\kappa := \min\{\mu_{\min}, 2\lambda \mu_{\min}/\mu_{\max}\}$.

Hence, by Grönwall's inequality and Equation (11), we have

$$\widehat{L}_\lambda(\theta(t)) \leq (\mu_{\max} + 3\mu_{\min}) \alpha^{2\epsilon} e^{-(t - T_2)/\kappa}. \tag{15}$$

For a contradiction, suppose $T_3 < \infty$, and let $i, i' \in I_+$ be such that $1 - \overline{w}_i(T_3)^\top \overline{w}_{i'}(T_3) = \alpha^{\epsilon/2}$. Then:

$$
\begin{aligned}
&1 - \overline{w}_i(T_3)^\top \overline{w}_{i'}(T_3) \\
&\leq 4\alpha^\epsilon - \int_{T_2}^{T_3} \frac{\mathrm{d}}{\mathrm{d}t} \overline{w}_i(t)^\top \overline{w}_{i'}(t)\, \mathrm{d}t &&\text{by Lemma 4 (b)} \\
&= 4\alpha^\epsilon - 2 \int_{T_2}^{T_3} \Big( \big( h_i(t) - \overline{w}_i(t) \overline{w}_i(t)^\top h_i(t) \big)^\top \overline{w}_{i'}(t) \\
&\qquad\qquad + \overline{w}_i(t)^\top \big( h_{i'}(t) - \overline{w}_{i'}(t) \overline{w}_{i'}(t)^\top h_{i'}(t) \big) \Big)\, \mathrm{d}t &&\text{by Equation (9)} \\
&= 4\alpha^\epsilon - 2 \int_{T_2}^{T_3} (1 - \overline{w}_i(t)^\top \overline{w}_{i'}(t)) \\
&\qquad\qquad \big( H(u_\star - v(t)) + \lambda \overline{u}_\star \big)^\top (\overline{w}_i(t) + \overline{w}_{i'}(t))\, \mathrm{d}t &&\text{by Equation (10)}
\end{aligned}
$$

$$\le 4\alpha^\epsilon - 2\int_{T_2}^{T_3} (1 - \overline{w}_i(t)^\top \overline{w}_{i'}(t))$$

$$H(u_\star - v(t))^\top(\overline{w}_i(t) + \overline{w}_{i'}(t))\,\mathrm{d}t \qquad\qquad \text{for small enough } \alpha$$

$$\le 4\alpha^\epsilon + 4\alpha^{\epsilon/2}\int_{T_2}^{T_3} \|H(u_\star - v(t))\|\,\mathrm{d}t \qquad\qquad \text{as } 1 - \overline{w}_i(t)^\top \overline{w}_{i'}(t) < \alpha^{\epsilon/2}$$

$$\le 4\alpha^\epsilon + 4\alpha^{\epsilon/2}\int_{T_2}^{T_3} \sqrt{\frac{\widehat{L}_\lambda(\theta(t))}{\mu_{\min}}}\,\mathrm{d}t \qquad\qquad \text{by Equation (14)}$$

$$\le 4\alpha^\epsilon + 4\alpha^{3\epsilon/2}\sqrt{\frac{\mu_{\max}}{\mu_{\min}} + 3}\int_{T_2}^{T_3} \exp\left(-\frac{t - T_2}{2\kappa}\right)\mathrm{d}t \qquad\qquad \text{by Equation (15)}$$

$$\le 4\alpha^\epsilon + 8\kappa\,\alpha^{3\epsilon/2}\sqrt{\frac{\mu_{\max}}{\mu_{\min}} + 3} \qquad\qquad \text{by integrating}$$

$$< \alpha^{\epsilon/2} \qquad\qquad \text{for small enough } \alpha,$$

which is a contradiction, and therefore $T_3 = \infty$, i.e., the $I_+$ neurons remain closely aligned at all times $t \ge T_2$.

Now, since by Equation (14) we have

$$\widehat{L}_\lambda(\theta(t)) = \|u_\star - v(t)\|_H^2 + \lambda\sum_{i\in I_+}\|w_i(t)\|^2\|\overline{u}_\star - \overline{w}_i(t)\|^2 + 2\lambda\sum_{i\in[m]\setminus I_+}\|w_i(t)\|^2, \qquad (16)$$

from Equation (15) we conclude that the composite vector $v(t)$ converges to $u_\star$, all the constituent neurons converge to perfect alignment, and all other neurons converge to 0:

$$\lim_{t\to\infty} v(t) = u_\star \qquad\qquad \forall i \in I_+, \lim_{t\to\infty}\overline{w}_i(t) = \overline{u}_\star \qquad\qquad \forall i \in [m]\setminus I_+, \lim_{t\to\infty} w_i(t) = 0.$$

Thus, to infer that $\lim_{t\to\infty}\theta(t) \in \Theta_{u_\star}$, it suffices to observe that for all $i \in I_+$ we have

$$|\mathrm{d}\|w_i(t)\|^2/\mathrm{d}t|$$

$$= 2\|w_i(t)\|^2\left|2\overline{w}_i(t)^\top H(u_\star - v(t)) - \lambda\|\overline{u}_\star - \overline{w}_i(t)\|^2\right| \qquad\qquad \text{by Equations (9) and (10)}$$

$$\le 5\|H(u_\star - v(t))\| + 2\lambda\|w_i(t)\|^2\|\overline{u}_\star - \overline{w}_i(t)\|^2 \qquad\qquad \text{for small enough } \alpha$$

$$\le 5\sqrt{\widehat{L}_\lambda(\theta(t))/\mu_{\min}} + 2\widehat{L}_\lambda(\theta(t)) \qquad\qquad \text{by Equation (16)}$$

$$\le \mathcal{O}(1)\exp\left(-\frac{t - T_2}{2\kappa}\right), \qquad\qquad \text{by Equation (15)}$$

so $\lim_{t\to\infty}\|w_i(t)\|^2$ exists. $\qquad\qquad\square$

Theorem 2 now follows by Proposition 3, fixing $\eta > 0$ and $\alpha^\star > 0$ so that Proposition 2 and Lemma 5 hold for all $\alpha \le \alpha^\star$ and $\lambda \le \mu_{\min}\alpha^{2\epsilon} = \mu_{\min}\alpha^\varepsilon$, and recalling that by Equation (7) we have $\|v_\star - u_\star\|_H^2 \le \|H(v_\star - u_\star)\|^2/\mu_{\min} = \lambda^2/\mu_{\min}$.

## D   Proof of Theorem 3

A main argument for the proof of Theorem 3 is the following key lemma.

**Lemma 6.** *If the data satisfies Assumption 3, and we have that* $m \ge 2$ *and* $0 < \lambda < \min\left(\sqrt{\sum_{k,y_k>0} y_k^2\|x_k\|^2}, \sqrt{\sum_{k,y_k<0} y_k^2\|x_k\|^2}\right)$, *then any global minimum* $(W^\star, a^\star) \in \mathbb{R}^{m\times(d+1)}$ *of either Equation* (Reg-$\lambda$) *or equation min-norm is such that there exist* $j_-, j_+ \in [m]$ *satisfying:*

$$\forall k \in [n], y_k < 0 \implies x_k^\top w_{j_+}^\star > 0,$$
$$\forall k \in [n], y_k > 0 \implies x_k^\top w_{j_-}^\star > 0.$$

In words, Lemma 6 states that any global minimum of either Equations (Reg-$\lambda$) and (min-norm) is such that it has one neuron $w_{j_+}$ that is positively correlated with all the training inputs corresponding to positive labels. Similarly, there is a neuron $w_{j_-}$ that is positively correlated to all the training inputs corresponding to negative labels. For the case of Equation (min-norm), this is a known fact that the global minima are equivalent to two such neurons (Boursier et al., 2022, Proposition 1). Lemma 6 extends it to the regularized problem equation Reg-$\lambda$. Lemma 6 is here weaker than Proposition 1 from Boursier et al. (2022), as it only gives a necessary condition for global minima. It is yet sufficient to our purpose, which is proving Theorem 3.

Similarly to the proof of Theorem 1, we introduce the notion of *weight cone*. For that, we define the following activation function which here omits the sign of the output neuron:

$$\tilde{A}_n : \quad \begin{matrix} \mathbb{R}^d \to \{0,1\}^n \\ w \mapsto (\mathbb{1}(w^\top x_k \geq 0)_{k \in [n]}) \end{matrix} \ .$$

Similarly, we associate to any binary vector $u \in \{0,1\}^n$ the weight cone $\tilde{\mathcal{C}}_u \subseteq \mathbb{R}^d$ as

$$\tilde{\mathcal{C}}_u = \tilde{A}_n^{-1}(u).$$

For any binary matrix $A \in \{0,1\}^{m \times (n+1)}$, we can decompose it in block as $A = \begin{bmatrix} A^0 A^1 \end{bmatrix}$ with $A^0 \in \{0,1\}^{m \times n}$ and $A^1 \in \{0,1\}^{m \times 1}$ such that activation and weight cones are related following

$$\mathcal{C}^A = \prod_{i=1}^m (\tilde{\mathcal{C}}_{A_i^0} \times \mathcal{R}_{A_i^1}),$$

where we define $\mathcal{R}_1 = \mathbb{R}_+$ and $\mathcal{R}_0 = \mathbb{R}_-^*$. In words, the parameters $(W, a) \in \mathbb{R}^{m \times (d+1)}$ belong to the activation cone $\mathcal{C}^A$ if and only if for any $i \in [m]$, the $i$-th weight belongs to the weight cone associated to the $i$-th row of $A$ and the sign of $a_i$ corresponds to the last element of such a row, i.e., $(\tilde{A}_n(w_i), \mathbb{1}(a_i \geq 0)) = A_i$.

From here, we can relate the fraction of non-empty cones containing global minima to the coupon collector problem. Indeed, if we choose uniformly at random an activation cone $\mathcal{C}^A$, it is equivalent to choosing independently, $m$ weight cones $(\tilde{\mathcal{C}}_{u_i})_{i \in [m]}$ and signs of output $\mathbb{1}(a_i \geq 0)$ uniformly at random – there is no empty activation cone in the case of orthogonal data.

Thus assuming $A$ is a binary matrix in $\{0,1\}^{m \times (n+1)}$, drawn uniformly at random among the set of binary matrices; and that the binary vectors $u_i \in \mathbb{R}^n$ are drawn i.i.d., uniformly at random among the set of binary vectors $u$, we have the following inequality, thanks to Lemma 6:

$$\mathbb{P}(\bar{\mathcal{C}}^A \text{ contains a global minimum of Equation (Reg-}\lambda\text{) or equation min-norm})$$
$$\leq \mathbb{P}(\exists j \in [m], \forall k \text{ s.t. } y_k > 0, u_{jk} = 1 \text{ and } \exists j \in [m], \forall k \text{ s.t. } y_k < 0, u_{jk} = 1)$$
$$\leq \min\left(\mathbb{P}(\exists j \in [m], \forall k \text{ s.t. } y_k > 0, u_{jk} = 1), \mathbb{P}(\exists j \in [m], \forall k \text{ s.t. } y_k > 0, u_{jk} = 1)\right).$$

Now note that in the orthogonal case, all weight cones are non-empty, i.e., choosing $u_i$ uniformly at random among the set of non-empty weight cones is the same as choosing its components $u_{ik}$ as independent Bernoulli variables of parameter $1/2$. As a consequence using a simple union bound,

$$\mathbb{P}(\exists j \in [m], \forall k \text{ s.t. } y_k > 0, u_{jk} = 1) \leq m 2^{-\#\{k \in [n] | y_k > 0\}}.$$

So that this finally yields when choosing $\mathcal{C}^A$ uniformly at random:

$$\mathbb{P}(\bar{\mathcal{C}}^A \text{ contains a global minimum of Equation (Reg-}\lambda\text{) or equation min-norm})$$
$$\leq m 2^{-\max(\#\{k \in [n] | y_k > 0\}, \#\{k \in [n] | y_k < 0\})}.$$

This finally yields the first part of Theorem 3.

From there, Lemma 7 below implies that the weight cones do not change during training in the presence of orthogonal data.

**Lemma 7.** *If the data satisfies Assumption 3, then for any $\lambda \geq 0$, the gradient flow solution $\theta$ of Equation* (3) *is such that for any $i \in [m]$ and $t \in \mathbb{R}$:*

$$\tilde{A}_n(w_i(0)) = \tilde{A}_n(w_i(t)).$$

*Proof.* This is a direct consequence of the following differential inclusion on any neuron $i \in [m]$ and data point $k \in [n]$:

$$\frac{\mathrm{d}(w_i(t)^\top x_k)}{\mathrm{d}t} = \dot{w}_i(t)^\top x_k$$

$$\in 2\frac{a_i(t)}{n}(y_k - f_{\theta(t)})\partial\sigma(w_i(t)^\top x_k)\|x_k\|^2 - 2\lambda w_i(t)^\top x_k.$$

The simplicity of the ODE comes from the orthogonality assumption. From there and using the expression of the subgradient of the ReLU activation, one can note that

$$w_i(t)^\top x_k < 0 \implies \frac{\mathrm{d}(w_i(t)^\top x_k)}{\mathrm{d}t} = -2\lambda w_i(t)^\top x_k.$$

This directly implies, using Grönwall's inequality, the fact that $w_i(t)^\top x_k < 0$ cannot become non-negative (in finite time) if it starts being negative. By continuity of $w_i(t)$, this also implies that it cannot become negative if it starts being non-negative. In other words, this implies that $\mathbb{1}(w_i(t)^\top x_k \geq 0)$ is constant during training. $\square$

As a consequence, any initialization $\theta(0)$ such that $\lim_{t\to\infty}\theta(t)$ exists satisfies for any $i \in [m]$ and $k \in [n]$,

$$w_i(0)^\top x_k \geq 0 \implies \lim_{t\to\infty} w_i(t)^\top x_k \geq 0$$
$$w_i(0)^\top x_k < 0 \implies \lim_{t\to\infty} w_i(t)^\top x_k \leq 0 \tag{17}$$

Equation (17) can be rewritten in a more compact way as

$$\forall i \in [m], \lim_{t\to\infty} w_i(t) \in \overline{\tilde{A}_n^{-1}(w_i(0))}.$$

As a consequence, any initialization that does not satisfy the second point of Theorem 3, i.e., any initialization such that $\lim_{t\to\infty}\theta(t)$ exists and is a global minimum is thus such that

$$\exists j \in [m], \forall k \text{ s.t. } y_k > 0, \tilde{A}_n(w_j(0))_k = 1,$$
$$\exists j \in [m], \forall k \text{ s.t. } y_k < 0, \tilde{A}_n(w_j(0))_k = 1. \tag{18}$$

Moreover by orthogonality of the data and rotation invariance of the initialization, the weight activations $\tilde{A}_n(w_i(0))$ are chosen uniformly at random at initialization among $\{0,1\}^n$. The property given by Equation (18) thus happens with probability at most

$$m2^{-\max(\#\{k\in[n]|y_k>0\},\#\{k\in[n]|y_k<0\})},$$

using the same argument as above. Since this property is necessary for the initialization to not satisfy the second point of Theorem 3, this allows to conclude.

## D.1 Proof of Lemma 6

Consider a global minimum $(W, a)$ of Equation (Reg-$\lambda$). First note that a stationary point $(W, a)$ of Equation (Reg-$\lambda$) is satisfies the following equality for any $i \in [m]$:

$$\frac{a_i}{n}\sum_{k=1}^n (f_\theta(x_k) - y_k)\partial\sigma(w_i^\top x_k)x_k + \lambda w_i \in 0.$$

The first equality implies

$$w_i \in \frac{a_i}{\lambda n} \sum_{k=1}^{n} (y_k - f_\theta(x_k)) \partial \sigma(w_i^\top x_k) x_k,$$

where $\partial \sigma(w_i^\top x_k) = \mathbb{1}(w_i^\top x_k > 0)$, except when $w_i^\top x_k = 0$. But then, the orthogonality directly implies that it indeed corresponds to the choice of subgradient 0, i.e.,

$$w_i = \frac{a_i}{\lambda n} \sum_{k=1}^{n} (y_k - f_\theta(x_k)) \mathbb{1}(w_i^\top x_k > 0) x_k.$$

Moreover, we know that any global minimum is balanced, i.e., $a_i^2 = \|w_i\|^2$. Denoting $\lambda_k = \frac{y_k - f_\theta(x_k)}{\lambda n}$, $B_i = \{k \in [n] \mid w_i^\top x_k > 0\}$ and $\mathsf{s}_i = \mathrm{sign}(a_i)$, the KKT condition finally rewrites for any $i \in [m]$:

$$w_i = \mathsf{s}_i \|w_i\| \sum_{k \in B_i} \lambda_k x_k.$$

In particular, for any $k \in [n]$,

$$w_i^\top x_k = \mathsf{s}_i \lambda_k \|w_i\| \|x_k\|^2 \mathbb{1}(w_i^\top x_k > 0),$$

i.e., if we define $A_k = \{i \in [m] \mid w_i^\top x_k > 0\}$, it implies that

$$A_k \subseteq \{i \in [m] \mid \mathsf{s}_i \lambda_k > 0\}. \tag{19}$$

Moreover, it also implies that for any $i$ and $k$,

$$w_i^\top x_k \leq 0 \implies w_i^\top x_k = 0. \tag{20}$$

From there, we can use a merging argument to show that the neural network $(W, a)$ is equivalent to the network $(\tilde{W}, \tilde{a})$ defined by

$$\forall i > 2, (\tilde{w}_i, \tilde{a}_i) = \mathbf{0}$$

$$\tilde{a}_1 = \sqrt{\left\| \sum_{i, \mathsf{s}_i=1} \|w_i\| w_i \right\|_2} \quad \text{and} \quad \tilde{a}_2 = -\sqrt{\left\| \sum_{i, \mathsf{s}_i=-1} \|w_i\| w_i \right\|_2}$$

$$\tilde{w}_1 = \frac{\sum_{i, \mathsf{s}_i=1} \|w_i\| w_i}{\tilde{a}_1} \quad \text{and} \quad \tilde{w}_2 = \frac{\sum_{i, \mathsf{s}_i=-1} \|w_i\| w_i}{\tilde{a}_2}.$$

It is indeed easy to check that for any $k \in [n]$, $f_\theta(x_k) = f_{\tilde{\theta}}(x_k)$. Moreover, we have by triangle inequality that

$$\frac{1}{2} \|\tilde{\theta}\|_2^2 = \|\tilde{w}_1\|_2^2 + \|\tilde{w}_2\|_2^2$$

$$\leq \sum_{i \in [m]} \|w_i\|_2^2$$

$$= \frac{1}{2} \|\theta\|_2^2.$$

By minimization of Equation (Reg-$\lambda$), the inequality is an equality. The equality case of triangle inequality then implies that

$$\forall i, j \in [m], \mathsf{s}_i = \mathsf{s}_j \neq 0 \implies \frac{w_i}{\|w_i\|} = \frac{w_j}{\|w_j\|}.$$

As a consequence, we can define

$$D_+ = \sum_{i, \mathsf{s}_i=1} \|w_i\| w_i \quad \text{and} \quad D_- = \sum_{i, \mathsf{s}_i=-1} \|w_i\| w_i.$$

Using Equation (20), we then have the following equality for any $k \in [n]$ :

$$
\begin{aligned}
f_\theta(x_k) &= \sum_{i,s_i=1} \|w_i\| \sigma(x_k^\top w_i) - \sum_{i,s_i=-1} \|w_i\| \sigma(x_k^\top w_i) \\
&= \sum_{i,s_i=1} \|w_i\| x_k^\top w_i - \sum_{i,s_i=-1} \|w_i\| x_k^\top w_i \\
&= D_+^\top x_k - D_-^\top x_k.
\end{aligned}
$$

Moreover, the above computations imply $D_+^\top x_k \geq 0$ and $D_-^\top x_k \geq 0$.

Additionally if $\lambda_k \geq 0$, Equation (19) implies that $A_k \subseteq \{i \in [m] \mid s_i = 1\}$. As a consequence if $\lambda_k \geq 0$,

$$
\begin{aligned}
f_\theta(x_k) &= \sum_{i \in A_k} a_i w_i^\top x_k \\
&= \sum_{i,s_i=1} a_i w_i^\top x_k \\
&= D_+^\top x_k,
\end{aligned}
\tag{21}
$$

so that if $\lambda_k \geq 0$, $f_\theta(x_k) \geq 0$, i.e.,

$$
\lambda_k \geq 0 \implies f_\theta(x_k) \geq 0.
\tag{22}
$$

We also have the following lemma.

**Lemma 8.** *If the data satisfies Assumption 3, and we have that $m \geq 2$ and $0 < \lambda \leq \min\left(\sqrt{\sum_{k,y_k>0} y_k^2 \|x_k\|^2}, \sqrt{\sum_{k,y_k<0} y_k^2 \|x_k\|^2}\right)$, then any global minimum $\theta^\star$ of Equation (Reg-$\lambda$) is such that for any $k \in [n]$,*

$$
y_k \neq 0 \implies f_{\theta^\star}(x_k) \neq 0.
$$

In particular, Equation (22) then becomes

$$
\lambda_k \geq 0 \implies f_\theta(x_k) > 0 \text{ or } y_k = 0.
$$

As by definition, $\lambda_k = \frac{y_k - f_\theta(x_k)}{\lambda n}$, this directly implies that

$$
\lambda_k \geq 0 \implies y_k \geq 0.
$$

But the symmetric argument also holds, i.e.,

$$
\lambda_k \leq 0 \implies y_k \leq 0.
$$

The converse leads to $y_k > 0 \implies \lambda_k > 0$. Equation (21) then implies that for any $k$ s.t. $y_k > 0$, $D_+^\top x_k > 0$. Moreover as $f_\theta(x_k) \neq 0$, there is at least one $i$ such that $s_i = 1$ and $w_i \neq \mathbf{0}$. Such a $w_i$ is then proportional to $D_+$ and thus corresponds to $w_{j_+}$ in Lemma 6.

Symmetrical arguments hold for the existence of $w_{j_-}$. $\qquad\square$

### D.2 Proof of Lemma 8.

Since $m \geq 2$, we can use the same merging argument as in Appendix D.1 to show that any global minimum is equivalent to a network with only two non-zero neurons. We can thus restrict here our analysis, without loss of generality, to a global minimum of Equation (Reg-$\lambda$) $\theta^\star = (W^\star, a^\star)$ such that

$$
\begin{aligned}
&\forall i > 2, (w_i^\star, a_i^\star) = \mathbf{0}, \\
&a_1^\star \geq 0 \quad \text{and} \quad a_2^\star \leq 0.
\end{aligned}
$$

Now assume by contradiction that $f_{\theta^\star}(x_k) = 0$ for some $k$ such that $y_k \neq 0$. Also assume without loss of generality that $y_k > 0$ – the negative case is dealt with symmetrically. Using the construction from Appendix D.1, Equations (19) and (20) imply in particular that $x_k^\top w_1^\star = 0$.

Suppose in a first case that $a_1^\star \neq 0$. From there for an arbitrarily small $\varepsilon > 0$, adding $\varepsilon x_k$ to $w_1^\star$ decreases the objective of Equation (Reg-$\lambda$).

Indeed, define $\theta^\varepsilon = (W^\varepsilon, a^\varepsilon)$ as:

$$\forall i \geq 2, (w_i^\varepsilon, a_i^\varepsilon) = (w_i^\star, a_i^\star),$$
$$(w_1^\varepsilon, a_1^\varepsilon) = (w_1^\star + \varepsilon x_k, a_1^\star).$$

Using the orthogonality between the $x_k$ then yields

$$L_\lambda(\theta^\varepsilon) = L_\lambda(\theta^\star) - 2(a_1^\star)\varepsilon y_k \|x_k\|^2 + (a_1^\star)^2 \varepsilon^2 \|x_k\|^4 + \lambda \varepsilon^2 \|x_k\|^2,$$

which is indeed smaller than $L_\lambda(\theta^\star)$ for a small enough $\varepsilon > 0$. This then contradicts the global minimality of $\theta^\star$.

Suppose now that $a_1^\star = 0$. In that case, $w_1^\star = \mathbf{0}$ by balancedness. We can then replace $\theta^\star$ by $\theta^\varepsilon$ where

$$\forall i \geq 2, (w_i^\varepsilon, a_i^\varepsilon) = (w_i^\star, a_i^\star),$$
$$a_1^\varepsilon = \varepsilon \|D_+\|,$$
$$w_1^\varepsilon = \varepsilon D_+,$$

were $D_+ = \sum_{k, y_k > 0} y_k x_k$. We can again compare the objectives as $\varepsilon \to 0$:

$$L_\lambda(\theta^\varepsilon) = L_\lambda(\theta^\star) - \frac{1}{n} \sum_{k, y_k > 0} 2\varepsilon^2 \|D_+\| \|x_k\|^2 y_k^2 + 2\lambda \varepsilon^2 \|D_+\|^2 + \mathcal{O}(\varepsilon^4),$$
$$= L_\lambda(\theta^\star) - \frac{2}{n} \varepsilon^2 \|D_+\|^3 + 2\lambda \varepsilon^2 \|D_+\|^2 + \mathcal{O}(\varepsilon^4).$$

The assumption on $\lambda$ then leads to a decrease of the loss, contradicting that $\theta^\star$ is a global minimum. This proves Lemma 8 by contradiction.

