# OpenReview forum: "Favorability of Loss Landscape with Regularization Requires Both Large Overparametrization and Initialization"
_TMLR — Accepted by TMLR_

### Review · Reviewer_LBtm · 2026-03-17

**Summary Of Contributions:**

This paper investigates the loss landscape and optimization dynamics of $l_2$-regularized two-layer ReLU networks. By analyzing activation cones, the authors theoretically prove that a favourable regularized landscape emerges under massive over-parametrization. Furthermore, using specifically constructed data examples, they highlight the difference in optimization trajectories between large and small initialization regimes. Overall, the paper establishes the theoretical level of over-parametrization required for a favourable regularized landscape.



**Strengths And Weaknesses:**

Strengths:

- It is nice that authors targeted the weight decay in the landscape perspective.
- The authors provide a  thorough theoretical analysis of the regularized loss landscape
- Discussion in the relationship between initialization and regularized landscape is interesting.

Weakness:

- In practice, evaluating whether the network reaches the unregularized global minima is as important as minimizing the regularized loss (Reg-$\lambda$), although the authors theoretically link the two as $\lambda \to 0$.  The paper would be even stronger if it included empirical or theoretical details on the extent to which the discovered minima satisfy the min-norm interpolation constraint.
- In Section 5, to connect Theorem 1 with large initialization, the authors present an interesting narrative based on the weight feature dynamic conjecture. To make this compelling point even more convincing, it would be a wonderful addition if the authors could include empirical observations in Appendix A.2 showing how the inner weights evolve during training.

**Audience:**

Yes

**Audience Explanation:**

This paper discusses the theoretical aspects of the loss landscape and weight decay. These findings will certainly be of interest to researchers working in related fields.

**Claims And Evidence:**

Yes

**Claims Explanation:**

Claims and evidences:

Claim1 (Favourable Landscape): Regularized landscape becomes favourable—i.e., spurious local minima represent a negligible fraction of local minima—under large overparametrization.

Evidence: Proved analytically in Theorem 1 and empirically validated in Section 7.

Claim2 (optimization dynamic under large initialization):  Regularized loss landscape results primarily hold relevance in the large initialization regime.

Evidence： Empirical results in A.2 (a little weak, see weakness 2)

Claim3 (optimization dynamic under large initialization):   for small initializations, optimization can still converge to spurious local minima.

Evidence: Proved analytically in Theorem 2 and empirically validated in Appendix A.2.

Claim4 (Necessity of Bound): the requirement of overparametrization  is indeed necessary for Theorem 1

Evidence: Proved analytically in Theorem 3

Overall, the primary claims in the paper are well-supported by theoretical evidence and illustrative experiments.

**Requested Changes:**

- see weakness.
- In Figures 7 and 8, the x-axis values decrease from left to right, which is a little strange.

---

> ### Author Response · Authors · 2026-04-21
>
> We thank the reviewer for their detailed and insightful review. In accordance to the reviewer's questions, we added some figures in the revised version.
>
> > In practice, evaluating whether the network reaches the unregularized global minima is as important as minimizing the regularized loss (Reg-$\lambda$), although the authors theoretically link the two as $\lambda\to 0$. The paper would be even stronger if it included empirical or theoretical details on the extent to which the discovered minima satisfy the min-norm interpolation constraint.
>
> Theorem 2 i) quantifies precisely how close is the achieved network to interpolation in the presence of regularization. We added in the revised version a figure that is similar to Figure 8, while plotting the achieved unregularized loss on the y axis (Figure 10 in revised version). It illustrates how the achieved solution nearly interpolates the data in practice.
>
> We are not sure to precisely understand the reviewer’s comment here, and we would be glad to provide any further experiments/discussion regarding it.
>
> > it would be a wonderful addition if the authors could include empirical observations in Appendix A.2 showing how the inner weights evolve during training.
>
> We thank the reviewer for this suggestion. Figure 9 of the original version was intended to illustrate this aspect. However, the quantity plotted on the y-axis might not be the best one to really observe this phenomenon. In the revised version, we modified this figure (Fig 11) and added a new figure (Fig 2) as well as a dedicated section in the main text (Section 7.2). The revised figure now has a readjusted x-axis starting from epoch 1, and we chose a different thresholding in the method (now showing all neurons that have norms $>10^{-4}$ when reweighted by output layer, instead of $10^{-6}$ in the old version), which better illustrates the difference between the two regimes. The new Figure 2 presents on the y-axis the effective rank of the weights matrix, which is a continuous proxy of the rank. These two figures now allow us to clearly distinguish between the lazy regime observed for large initializations and the rich regime for small initializations. We also added another figure (Figure 12 in revised version) that shows the evolution of (unregularized loss) along training. This allows to show that (near) interpolation is quickly reached, highlighting the late grokking phase that can then be observed from Figures 2 and 11 (see below). Precisely, we observe with this new version the following:
>
> - *large init:* both number of neuron directions and effective rank are high (5 being full rank for the effective rank) at the first epochs and remain more or less constant until epoch $10^7$, corresponding to the lazy regime. At this point, both values will decrease similarly to the old version figure, due to a late grokking phenomenon.
>
> - *small init:* effective rank is high at initialization, but quickly becomes small, which corresponds to the feature learning regime. On the other hand, the number of neuron directions starts at 0 (due to the looser threshold), first increases and then gets to a small value after a few epochs, when near interpolation of the data is already reached (see Figure 12). The late grokking phase then does not change much about both values, slightly decreasing them further.

---

### Review · Reviewer_6o9v · 2026-03-20

**Summary Of Contributions:**

This paper investigates the optimization of two-layer ReLU networks with $l_2$ regularization (weight decay). The authors establish that almost all  non-empty active cones contain a global minimum and lack spurious local minima with width of $m \gtrsim  \min(n^d, 2^n)$. The paper demonstrates that this bound is both sufficient (Theorem 1) and necessary, with the necessity proven via an analysis of orthogonal data (Theorem 3). Also, the authors investigate the optimization dynamics. With small initialization regime and small regularization parameter, subgradient flow may converge to spurious local minima that represent sparse interpolators, despite the landscape's theoretical favourability.

**Audience:**

Yes

**Audience Explanation:**

This paper focuses on the optimization of two-layer neural networks, extending previous theoretical results to weight decay setting. Given the fundamentally different and novel conclusions drawn in this setting, the paper will appeal to researchers within the learning theory community.

**Claims And Evidence:**

Yes

**Claims Explanation:**

I have checked most of the proofs and did not identify any issues. Furthermore, the experimental results in the paper consistently match the theoretical findings.

**Requested Changes:**

1. The "weight decay" in this paper refers to an $L_2$ regularizer. However, in modern optimizers (e.g., Adam, Muon), weight decay is not equivalent to $L_2$ regularization. Hence, I think it is vital to clarify these definitions.

2. Theorems 1 and 3 suggest that the loss landscape with weight decay is worse than the landscape without it. However, from an $L_2$ regularization perspective, the term $\lambda \|\theta\|^2_2$ typically improves the convexity of the loss function, which should improve the loss landscape. Could you provide some explanations or insights regarding this mismatch?

3. Immediately following Theorem 1, the text states: "In other words, for $N$ the number of non-empty activation cones, Theorem 1 states that at least $(1 - \varepsilon)N$ cones satisfy items (i) and (ii) above." However, this statement seems to depend on the specific sampling method used.

4. In the proof of Theorem 1 (Page 21), there appears a statement $A_n(w^\star_i,a^\star_i) \neq A_n(w^\star_j,a^\star_i)$. I think this needs some detailed explanation.

---

> ### Author Response · Authors · 2026-04-21
>
> We thank the reviewer for their detailed and insightful review.
>
> > 1. The "weight decay" in this paper refers to an $L_2$ regularizer
>
> Indeed, we agree with the reviewer that weight decay and $L_2$ regularization do not coincide with modern optimizers as Adam and Muon. We clarified these two notions in the introduction of the revised version, as well as changed the title in consequence to “Favorability of Loss Landscape with Regularization Requires Both Large Overparametrization and Initialization”.
>
>
> > 2. Theorems 1 and 3 suggest that the loss landscape with weight decay is worse than the landscape without it
>
> It depends on what “worse” means for a loss landscape. Indeed, adding a $\lambda |\theta|^2$ should make the landscape more convex, which is typically associated with “easier to optimize”. However, due to the original loss landscape, the “number” (although infinite in both cases) of global minima is much smaller once we add weight regularization in contrast to the unregularized case. This is because the unregularized case contains large flat regions corresponding to the global minima (ie the interpolation manifold), while the addition of a regularization term makes these regions not flat anymore, with only a few points now being global minima.
>
> This seemingly weird result is actually quite natural: without regularization, there are many different ways to interpolate the data; however there are only a few–if not unique–minimal ways to interpolate this data, making them much harder to find. However, we’d like to insist that we do not claim that minimizing with regularization is a worse choice here: even if we do not manage to find a global minimum of the regularized objective, we believe the found solution would still present better regularities than a (global) minimum that would typically be found on the unregularized objective. We discuss this after Theorem 1 in the revised version
>
> > 3. "In other words, for N the number of non-empty activation cones, Theorem 1 states that at least (1-\varepsilon)N cones satisfy items (i) and (ii) above." However, this statement seems to depend on the specific sampling method used.
>
> This sentence (and also the Theorem) does not refer to any sampling procedure: the number of cones is finite and denoted by $N$. As pointed out by **Reviewer CCsX** however, Theorem 1 is formulated in terms of the number of non-empty cones rather than weight measure, which may be a more relevant notion for typical initialization schemes. We refer to our response to **Reviewer CCsX** for a more detailed discussion of this distinction.
>
> > 4. In the proof of Theorem 1 (Page 21)
>
> We thank the reviewer for raising this statement, which appears to have a typo, it should be $A_n(w_i^\star, a_i^\star)\neq A_n(w_j^\star, a_j^\star)$. This statement means that we can choose the global minimum such that inside a non-empty neuron activation cone, it has a single non-zero neuron. This derives from Wang et al (2022, Proposition 2) which states that if we have several non-zero neurons within the same cone, we can merge them into a single neuron, for which the regularized loss is smaller. We added an explanation after the equation in the revised version.

---

### Review · Reviewer_CCsX · 2026-04-10

**Summary Of Contributions:**

The paper investigates the loss landscape of two-layer ReLU neural networks with regularized loss. Specifically, they showed the following results:

1. Under ultra-parameterization, most activation cones contain a global minimum but not spurious minimum.
2. With a specific data setting, the gradient descent trajectory leads to a spurious local minimum.
3. Without ultra-parameterization, spurious minimum can be common with specific data settings.

Overall the paper is technically sound with refined characterization of the loss landscape and the gradient dynamics for the two-layer ReLU networks. However, some technical proof details are not very clear and need additional clarification.

Pros:
1. Most results are technically sound.
2. Technical proofs sufficiently utilize the homogeneity of the ReLU activation as well as the discontinuity of sign function.
3. Qualitatively, experiment results match with the theoretical results.

Cons:
1. Some proof details are not too clear and need further clarification or correction.
2. Assumptions of both positive and negative results are a bit strong.

Specific issues are elaborated in the sections below.

**Audience:**

Yes

**Audience Explanation:**

The paper investigates the loss landscape and the training dynamics of a regularized two-layer ReLU network in detail. For people interested in the training dynamics of ReLU networks, this paper is definitely of value.

Indeed the assumptions of the results are strong and a bit far from practical. The $n^d$ or $2^n$ width is impractical for real training tasks in Theorem 1. In the negative results such as Theorem 2 and 3, specific assumptions are imposed for data, which may also hardly hold in practice. Nevertheless, these assumptions are understandable for a paper with mainly theoretical purposes, and experiments do show a trend that width improves the loss landscape.

I do have one question about the scope of Theorem 1. Theorem 1 claims that most of the activation cones contain global minimum and no spurious minima. But these cones are not directly defined by weights $(W, a)$, but a combination of them. Therefore, it seems that some cones may contain "a large measure" of weights than the others, and that the activation cones should not be treated with equal probability. Then the question is: is it possible that most of the weights in the parameter space actually fall into the activation cones that are not good enough? Can we directly measure the probability of spurious local minima in the weight measure?

**Broader Impact Concerns:**

This paper is mainly a theoretical paper and I do not have concerns on broader impact statements.

**Claims And Evidence:**

Yes

**Claims Explanation:**

In general, most theoretical claims are correct with detailed proofs. Most experiment results match the theoretical results in trends. There are unclear points, mostly in theoretical proofs, listed as below:

1. Line 5, page 30: In the proof of Lemma 5, Lemma 3b is cited in the beginning to establish properties for $i \in I_+$. However, Lemma 3b only covers $i \in [m] \backslash I_+$. The correct lemma should be specified here.
2. Page 31: the last inequality is not too clear to me. It's not too clear why decomposing $v(t)$ as $\bar{u}^\star (t) \bar{u}^\star (t)^\top v(t)$ and $v(t)-\bar{u}^\star(t) \bar{u}^\star (t)^\top v(t)$ leads to the second term in the right hand side.
3. Page 33 Line 1: In the right hand side, the expansion should be $2v^\star -v(t)-u^\star$ instead of $v^\star-u^\star$? If so then the remaining inequalities in Equation (13) may not hold.

Additional typos:

1. Proof of Proposition 1c: the coefficients should be $9/8$ and $9/2$ instead of $9/(4\sqrt{2})$ and $9/\sqrt{2}$.
2. Proof of Lemma 6, line 6: $\sigma(w_i^\top x_k)$ should be $\partial \sigma(w_i^\top x_k)$.
3. Page 37, line 1: The second $\tilde{a}_1$ should be $\tilde{a}_2$.

**Requested Changes:**

Please refer to the sections above.

---

> ### Author Response · Authors · 2026-04-21
>
> We thank the reviewer for their detailed review and careful reading of our proofs.
>
> > 1. Line 5, page 30
>
> We have expanded this paragraph, and clarified the application of Lemma 3b, whose role is to ensure that the neurons with indices outside $I_+$ do not contribute to the network outputs $f_{\theta(t)}(x_k)$ (since, starting at time $T_1$, they remain deactivated from all training points).
>
> > 2. Page 31
>
> We have added a paragraph to justify this inequality in greater detail.
>
> > 3. Page 33 Line 1
>
> It is $v^* - v(t)$ because of the cancellation of $u*$ and $-u*$.  We have expanded the justification in the margin to clarify this.
>
> > Additional typos
>
> We thank the reviewer for catching these typos that will be corrected accordingly.
>
> > the assumptions of the results are strong and a bit far from practical.
>
> Indeed, the overparametrization level required by Theorem 1 might not be desirable in practice. Note that Theorem 3 illustrates that this assumption is yet necessary from a general aspect, and can thus be seen as some “negative” results in the sense that we really need a large number of parameters to have this favorable landscape (as also observed empirically). Our belief is that in practice, even though we do not necessarily converge towards a global minimum of the regularized problem, we might still converge towards a local minimum that is reasonably good and still yields nice generalization.
>
> We extended the comment about that in the last paragraph of the Conclusion.
>
> > it seems that some cones may contain "a large measure" of weights than the others, and that the activation cones should not be treated with equal probability
>
> Indeed, this is an important subtlety in our work: here, we quantify our theorem in terms of non-empty cones, as having such results in terms of weight measure is more intricate, and might require additional assumptions on the data distribution.  This subtlety is discussed more clearly in the revised version, after Theorem 1.
>
> While our main experiments sample the cones uniformly at random (to directly illustrate our Theorem 1), Appendix A.1 (see **Alternative cone sampling procedures and datasets** paragraph) considers an alternative sampling method given by initializing the network weights according to a standard Gaussian – which would then be representative of Theorem 1 when counting in terms of measures of the cones. While the experiments yield similar results for Figure 3, 4 and 6, the conclusion is different for Figure 5, where the landscape seems even more favorable.

---

### Author Response · Authors · 2026-04-21

We thank the reviewers for their very detailed and insightful feedback. We answer their questions in specific comments below. We also uploaded a revised version of the paper that incorporates all the reviewers’ comments – we detail in our rebuttals the specific parts of the paper where we extended/modified the previous version.

---

### Decision · Action_Editor_dGK2 · 2026-06-03

**Recommendation:** Accept as is

**Additional Comments:**

The reviewers' concerns were satisfactorily addressed in the revision, and the reviewing team is positive about the contribution. The paper provides a thorough theoretical investigation of regularized two-layer ReLU networks, combining favorable landscape guarantees, necessity results, and optimization-dynamics analysis. The reviewers found the theoretical claims to be well supported and the experimental results consistent with the theory. The presentation of the paper was also improved through the revision process.

The reviewing team believes that the paper meets the standards for publication in TMLR.

**Audience:**

Yes

**Audience Explanation:**

The paper addresses fundamental questions concerning optimization and generalization in neural network training with regularization. The loss landscape of overparameterized neural networks, the role of regularization, and the interaction between initialization and optimization dynamics are topics of ongoing interest in optimization theory and learning theory. The reviewing team agrees that the results will be of interest to researchers working on the theoretical foundations of machine learning, neural network optimization, and loss landscape analysis.

**Claims And Evidence:**

Yes

**Claims Explanation:**

This work studies the loss landscape and optimization dynamics of \ell_2-regularized two-layer ReLU networks. The paper establishes that, under a sufficiently large level of overparameterization, almost all non-empty activation cones contain a global minimum and no spurious local minima. The authors further show that this level of overparameterization is not only sufficient but also necessary through an analysis of orthogonal data. In addition, the paper investigates the role of initialization and demonstrates that favorable landscape properties do not necessarily translate into successful optimization outcomes in the small-initialization regime, where optimization may still converge to spurious local minima.

The reviewers agree that the theoretical results are technically sound and supported by rigorous analysis. The paper provides both positive and negative results, including sufficiency, necessity, and optimization-dynamics characterizations, resulting in a comprehensive treatment of the problem. The empirical results are consistent with the theoretical findings and help illustrate the practical implications of the analysis. The revision and author responses addressed the main reviewer concerns regarding technical details, interpretation of the landscape results, and the relationship between initialization and optimization. The subsequent recommendations were positive, with reviewers agreeing that the claims are supported by clear evidence.

Overall, the reviewing team agrees that the claims are supported by accurate, convincing, and clear evidence.